# Human CST suppresses origin licensing and promotes AND-1/Ctf4 chromatin association

Yilin Wang[1,*], Kathryn S Brady[1,*], Benjamin P Caiello[1], Stephanie M Ackerson[1], Jason A Stewart[1,2]

**Human CTC1-STN1-TEN1 (CST) is an RPA-like single-stranded DNA-binding protein that interacts with DNA polymerase $\alpha$-primase (pol $\alpha$) and functions in telomere replication. Previous studies suggest that CST also promotes replication restart after fork stalling. However, the precise role of CST in genome-wide replication remains unclear. In this study, we sought to understand whether CST alters origin licensing and activation. Replication origins are licensed by loading of the minichromosome maintenance 2–7 (MCM) complex in G1 followed by replisome assembly and origin firing in S-phase. We find that CST directly interacts with the MCM complex and disrupts binding of CDT1 to MCM, leading to decreased origin licensing. We also show that CST enhances replisome assembly by promoting AND-1/pol $\alpha$ chromatin association. Moreover, these interactions are not dependent on exogenous replication stress, suggesting that CST acts as a specialized replication factor during normal replication. Overall, our findings implicate CST as a novel regulator of origin licensing and replisome assembly/fork progression through interactions with MCM, AND-1, and pol $\alpha$.**

## Introduction

DNA replication must occur with high fidelity and efficiency to preserve genome stability. Each time human cells divide, 50,000–100,000 DNA replication origins are activated for genome duplication (Zhai et al, 2017b; Higa et al, 2017; Riera et al, 2017). During telophase and G1, replication origins are licensed by binding of the origin recognition complex (ORC) and CDC6 to the DNA, followed by recruitment of CDT1 and minichromosome maintenance 2–7 (MCM). Loading of the first MCM hexamer by ORC and CDC6 leads to the formation of ORC-CDC6-CDT1-MCM (OCCM) complex. A second MCM hexamer is then recruited and loaded onto the DNA to form the pre-replication complex (pre-RC). Recruitment and loading of MCMs are dependent on CDT1. CDT1 facilitates interaction between MCM and ORC-CDC6 and also stabilizes opening of the MCM hexamer for loading onto the DNA (Masai et al, 2010;

Pozo & Cook, 2016; Frigola et al, 2017). Once the first MCM hexamer is loaded, CDT1 and CDC6 are released. A second MCM–CDT1 complex along with CDC6 then binds ORC, leading to loading of a second MCM hexamer (Ticau, 2015, 2017). Loading of the two MCM hexamers constitutes a licensed replication origin. Origin licensing is restricted to telophase and G1 of the cell cycle to prevent re-replication in S-phase. Unlike budding yeast, origin licensing in mammals is not defined by DNA sequence but by chromatin context and accessibility (Cayrou et al, 2015).

Upon entering S-phase, replication factors are recruited to origins to form the pre-initiation complex (pre-IC). MCM is bound by CDC45 and GINS to form the CDC45–MCM–GINS (CMG) complex, which serves as the replicative helicase (Deegan & Diffley, 2016). Three DNA polymerases (pol) are recruited during replisome assembly and used for DNA synthesis upon origin firing. Pol $\varepsilon$ binds directly to CMG, whereas pol $\delta$ and pol $\alpha$-primase (pol $\alpha$) are linked to the replisome by PCNA and Ctf4/AND-1, respectively. Once assembled, the replisome is then activated, or fired, after phosphorylation of MCM by Dbf4-dependent kinase and cyclin-dependent kinase. Origin licensing and activation was recently reconstituted with purified replication factors from budding yeast (Yeeles et al, 2015). However, many questions remain, particularly in regards to where replication origins are licensed in higher eukaryotes and how they are selected for activation. Here, we identify human CTC1-STN1-TEN1 (CST) as a novel regulator of origin licensing and replisome assembly. CST is an RPA-like single-stranded (ss)DNA-binding protein that has primarily been characterized as a telomere replication factor with less well-understood roles in genome-wide replication (Stewart et al, 2018).

Our previous work indicated that CST promotes origin firing in response to genome-wide replication stress (Stewart et al, 2012). In addition, work by Chastain et al showed that CST recruits RAD51 to rescue stalled replication and prevent chromosome fragility at GC-rich DNA (Chastain et al, 2016). However, the mechanism by which CST facilitates replication restart remains unclear. CTC1 and STN1 were originally discovered as pol $\alpha$ accessory factors (Goulian et al, 1990; Casteel et al, 2009). CST stimulates pol $\alpha$-primase activity and the primase-to-polymerase switch (Nakaoka et al, 2012; Ganduri & Lue, 2017). Nevertheless, CST does not localize to active replication

[1]Department of Biological Sciences, University of South Carolina, Columbia, SC, USA   [2]Center for Colon Cancer Research, University of South Carolina, Columbia, SC, USA

Correspondence: jason.stewart@sc.edu
*Yilin Wang and Kathryn S Brady contributed equally to this work

forks, suggesting it may function before replication initiation and/ or at stalled replication forks (Miyake et al, 2009; Sirbu et al, 2013). Moreover, stable depletion of CST subunits did not alter bulk DNA replication in HeLa cells under normal conditions but does result in increased anaphase bridges and chromosome fragility, suggesting that CST is likely used at specific regions of the genome (Stewart et al, 2012; Wang et al, 2012; Chastain et al, 2016; Wang & Chai, 2018).

In agreement with this idea, in vitro biochemical analysis revealed that CST binds and resolves G-quadruplexes (G4s) (Bhattacharjee et al, 2017). Chromatin-immunoprecipitation with sequencing analysis also demonstrated that STN1 localizes to non-telomeric GC-rich regions, which are known to form G4s (Chastain et al, 2016). G4s are stable, four-stranded structures that can block replication, regulate RNA transcription, and are associated with several diseases (Maizels, 2015; Rhodes & Lipps, 2015). G4s are also enriched at DNA replication origins and may promote origin licensing (Valton & Prioleau, 2016).

During telomere replication, CST participates in many of the steps required for telomere maintenance. These steps include replication of the telomere duplex, removal of telomerase, prevention of ATR activation, and engagement of pol α for C-strand fill-in synthesis (Miyake et al, 2009; Surovtseva et al, 2009; Chen et al, 2012; Gu et al, 2012; Stewart et al, 2012; Wang et al, 2012; Gu & Chang, 2013; Kasbek et al, 2013; Feng, et al, 2017, 2018). CST was also recently shown to interact with the shieldin complex to counteract double-strand break (DSB) end resection by facilitating fill-in by pol α, similar to its role in telomeric C-strand fill-in (Barazas et al, 2018; Mirman et al, 2018). This multi-functionality of CST in telomere maintenance, DSB repair, and replication rescue is not unexpected given its similarity to RPA.

RPA directs various transactions during DNA replication and repair through the use of multiple oligonucleotide–oligosaccharide binding (OB)-folds, which allow RPA to bind ssDNA of different lengths and configurations (e.g., ss-dsDNA junctions) (Fanning et al, 2006; Chen & Wold, 2014). Like RPA, CST is composed of multiple OB-folds, which allow it to bind different forms of ssDNA (Miyake et al, 2009; Chen et al, 2012; Bhattacharjee et al, 2017). Although CST and RPA have similar binding affinities, CST, unlike RPA, is in low abundance and may require recruitment to specific sites. For example, recruitment of CST to DSBs is dependent on the shieldin complex, whereas TPP1 appears to localize CST to telomeres (Chen et al, 2012; Mirman et al, 2018). Use of CST may be advantageous because, unlike RPA, its binding is not expected to elicit ATR activation (Feng et al, 2017).

Mutations in CTC1 and STN1 cause two pleiotropic autosomal recessive disorders, Coats plus and dyskeratosis congenita (Anderson et al, 2012; Keller et al, 2012; Polvi et al, 2012; Simon et al, 2016). Dyskeratosis congenita is caused by accelerated telomere shortening, which leads to cell proliferation defects and ultimately bone marrow failure (Armanios & Blackburn, 2012; Stanley & Armanios, 2015). Coats plus shares features of dyskeratosis congenita but has additional features, including intercranial calcifications, retinopathy, intrauterine growth retardation, and gastrointestinal bleeding (Briggs et al, 2008). Interestingly, some Coats plus patients do not exhibit accelerated telomere shortening, suggesting that both telomere and non-telomere defects contribute to the disease (Polvi et al, 2012; Romaniello et al,

2013). Thus, it is important to determine the different roles of CST in preserving genome stability to characterize the molecular etiology of these diseases.

Here, we present evidence that CST interacts with the MCM complex and suppresses origin licensing by disrupting the interaction between MCM and CDT1. Furthermore, we find that CST is important for recruitment/stabilization of AND-1 and pol α on the chromatin. Interestingly, regulation of both origin licensing and AND-1/pol α chromatin association occur in the absence of hydroxyurea (HU)-induced replication fork stalling, suggesting that these functions are independent of CST's role in origin activation after exogenous replication stress. Together, our results support a direct role for CST in both origin licensing and replication activation.

## Results

### STN1 depletion leads to increased levels of chromatin-bound MCM

Because the precise role(s) of CST in general replication are poorly understood, we tested whether CST affects the levels of chromatin-bound MCM in the absence of replication stress, as this would suggest changes in origin licensing and activation. To determine whether depletion of STN1 alters MCM levels, we used previously characterized HeLa1.2.11 cells with stable shRNA STN1 knockdown (shSTN1) to measure the levels of chromatin-bound MCM (Fig 1) (Stewart et al, 2012). Cells expressing a non-targeting shRNA (shNT) or shSTN1 cells expressing an shRNA resistant Flag-STN1 (shSTN1-Res) were used as controls (Fig 1A). 5-Ethynyl-2´-deoxyuridine (EdU) was added to the cultures 30 min prior to sample processing, which enabled later identification of S-phase cells (see Fig 2A). The cells were then harvested, pre-extracted to remove soluble MCM, fixed, and examined by indirect immunofluorescence (IF) to detect chromatin-bound MCM7, MCM3, or MCM6 (Fig 1B and C). MCM fluorescence intensity was measured in each nucleus and the MCM signal intensity compared between cell lines (Fig 1C). The analysis revealed that the levels of chromatin-bound MCM7, MCM3, and MCM6 were significantly increased in shSTN1 cells compared with controls (Fig 1C). These results were confirmed in HCT116 shSTN1 cells for chromatin-bound MCM6 (Fig S1A and B). The increase is specific to chromatin-bound MCM, as total cellular MCM3, MCM6, or MCM7 levels were similar in shSTN1 and controls cells (Fig S1C). These findings indicate that depletion of STN1 significantly increases chromatin-bound MCM.

### CST overexpression leads to decreased chromatin-bound MCM

We next tested whether CST overexpression (CST-OE) also affected MCM chromatin association. For this experiment, we used a previously described HeLa cell line overexpressing all three CST subunits (Fig 1D) (Wang et al, 2014). IF was performed, as described above, for either MCM7 (Fig 1E and F) or MCM6 (Fig S1D). CST-OE had the reverse effect to STN1 depletion as it led to a substantial decrease in chromatin-bound MCM. Together, the results in Fig 1 indicate that the level of chromatin-bound MCM is inversely

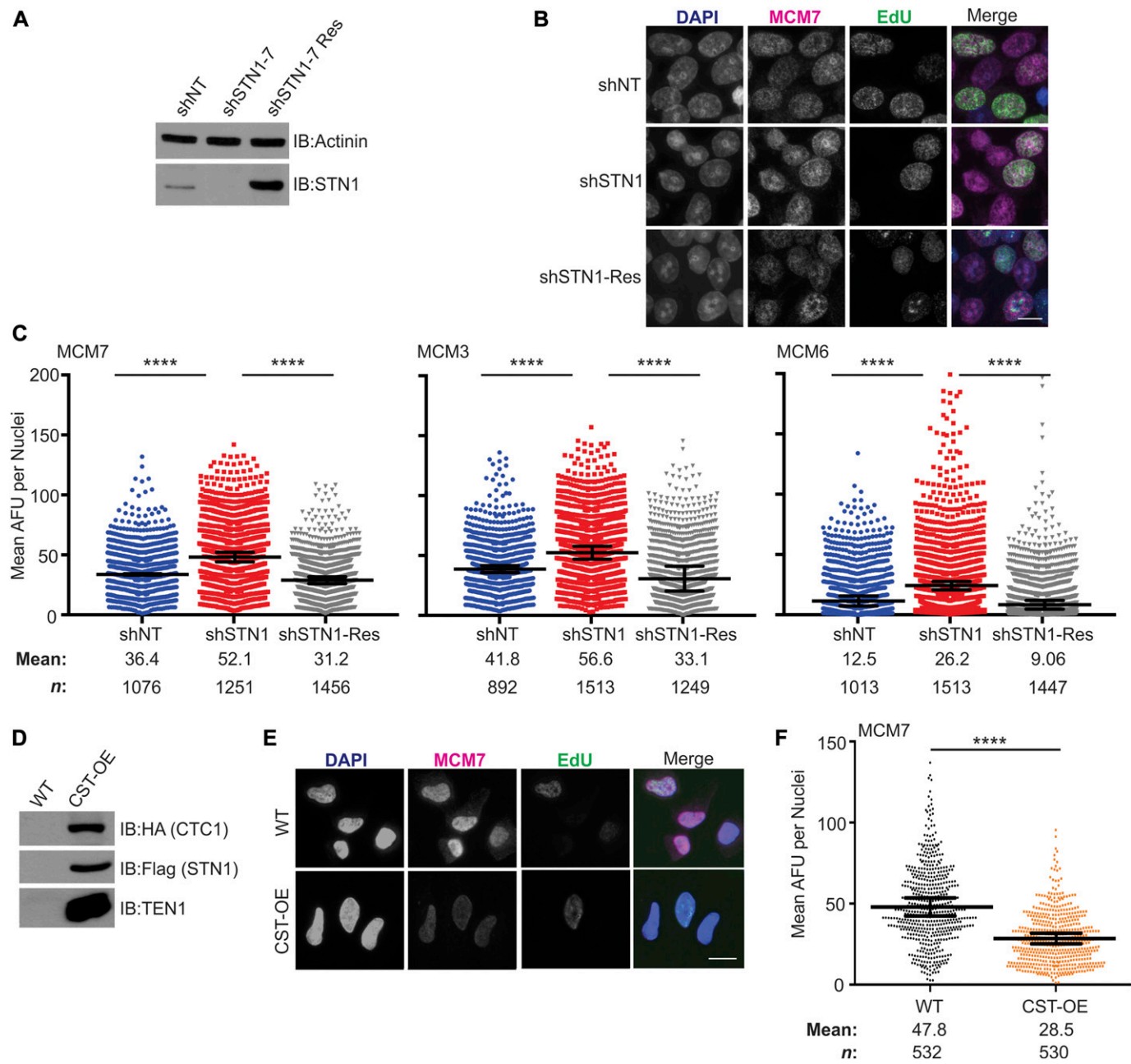

**Figure 1. Altered CST expression leads to changes in chromatin-bound MCM.**
**(A)** Western blot analysis of STN1 knockdown in HeLa cells. Actinin is used as a loading control. shNT: non-targeting shRNA; shSTN1: STN1 shRNA; and shSTN1-Res: shSTN1 cells plus shRNA-resistant Flag-STN1. **(B)** Representative images of pre-extracted, EdU-lableled cells used to measure MCM levels. DAPI: blue, MCM: magenta, EdU: green. Scale bar = 12.5 µm. **(C)** Dot plots of mean MCM7, MCM3, or MCM6 intensity per nuclei, represented in arbitrary fluorescent units (AFU). Black line and numbers below the graph indicate the mean AFU. Error bars indicate the ±SEM of at least three independent biological experiments. n indicates the number of total nuclei scored. **(D)** Western blot analysis of HA-CTC1, Flag-STN1, and untagged TEN1 in CST overexpressing (CST-OE) and WT cells. **(E)** Representative images of HeLa WT and CST-OE cells, as in (B). Scale bar = 12.5 µm. **(F)** Dot plots of mean MCM7 intensity per nuclei. Black line and numbers below the graph indicate the mean AFU. Error bars indicate the ±SEM of three independent biological experiments. n indicates the number of total nuclei scored. P-values were calculated by an unpaired, two-tailed Mann–Whitney test (****$P \leq 0.0001$).

proportional to CST or STN1 expression. It is notable that the changes in MCM occur in the absence of exogenous replication stress, which indicates that CST regulates MCM under normal conditions rather than in response to genome-wide replication fork stalling or DNA damage.

**STN1 depletion leads to increased MCM in G1 and S-phase**

We next wanted to determine whether the changes in chromatin-bound MCM were due to alterations in loading of MCM in G1 or unloading in S/G2. MCM loading onto the chromatin is tightly

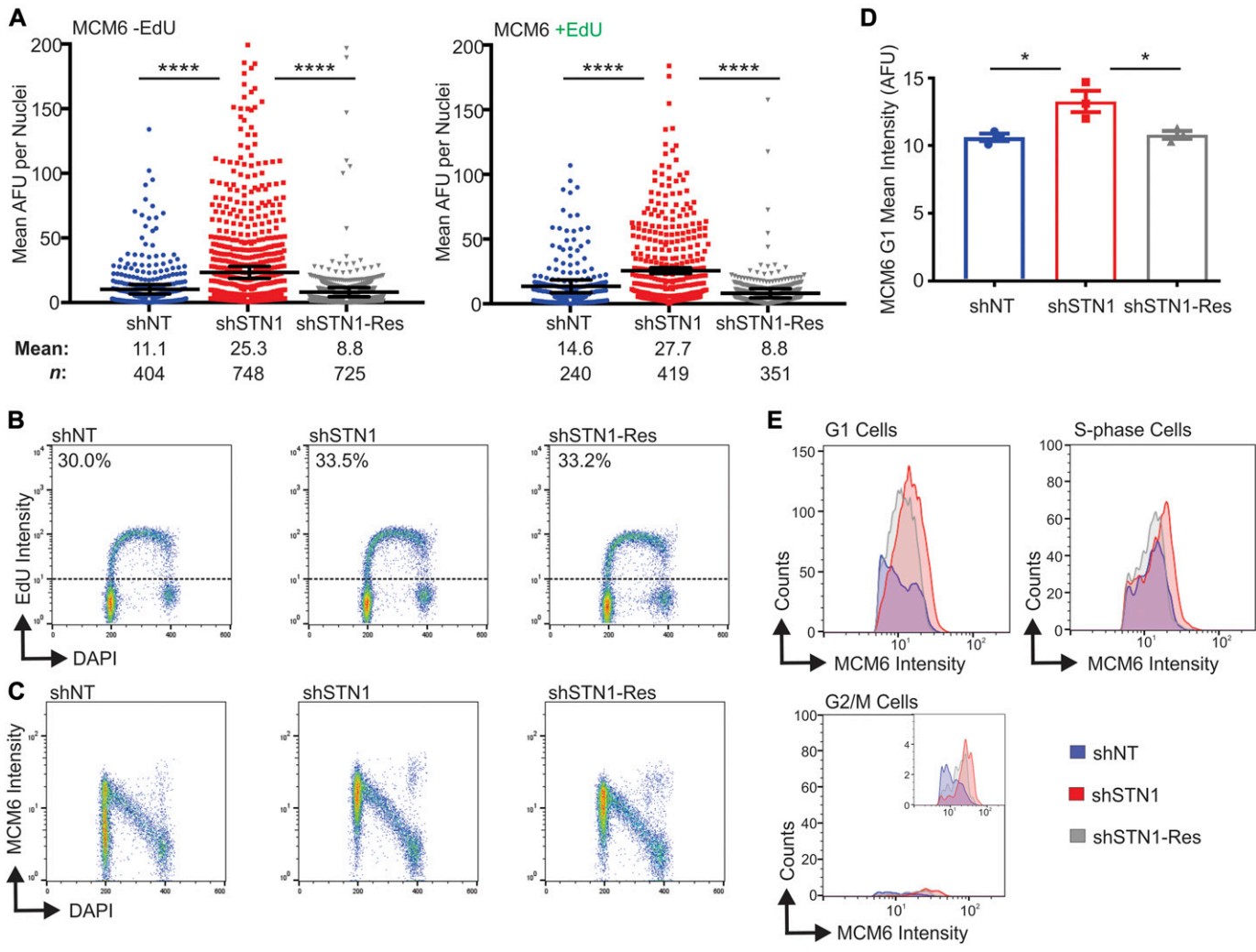

**Figure 2. Chromatin-bound MCM increases in both G1 and S-phase with STN1 depletion.**
The indicated cell lines were labeled with EdU, pre-extracted, and MCM6 detected. **(A)** Dot plots of mean MCM6 intensity. S-phase cells are indicated as +EdU. Black line and numbers below the graph indicate the mean AFU. Error bars indicate the ±SEM of three independent biological experiments. *n* indicates the number of total nuclei scored. **(B–E)** Flow cytometry data are representative of four independent biological experiments. **(B)** DNA content (DAPI) versus EdU signal intensity. Numbers above the dashed line represents the percentage of EdU+ cells. **(C)** DNA content versus MCM6 intensity. **(D)** Graph of the mean intensity of G1 MCM6-positive cells (see the Materials and Methods section and Fig S2 for gating). **(E)** Histograms showing the distribution of MCM signal intensity for G1, S-phase, or G2/M cells (see the Materials and Methods section). *P*-values were calculated by an unpaired, two-tailed Mann–Whitney test in **(A)** and *t* test in **(D)** (****P ≤ 0.0001, *P ≤ 0.05).

regulated throughout the cell cycle with origin licensing restricted to telophase and G1. MCM is then unloaded from the chromatin after DNA synthesis in a ubiquitin-dependent manner (Moreno et al, 2014; Dewar et al, 2015; Dewar et al, 2017; Sonneville et al, 2017). In initial experiments, we determined the effect of STN1 knockdown on the levels of chromatin-bound MCM in EdU positive versus negative cells (Figs 1B and 2A). This allowed us to distinguish whether the increase was restricted to S-phase. We found that MCM6 levels significantly increased in both EdU-positive and EdU-negative shSTN1 cells, indicating that the increase in MCM occurs both outside and within S-phase (Fig 2A).

Because the above experiments did not distinguish between cells in G1 versus G2, we next performed flow cytometry to separate MCM-positive cells in G1, S, and G2/M. This allowed us to address whether the changes in chromatin-bound MCM reflected increased

MCM loading (i.e., origin licensing) in G1 versus MCM dissociation in S/G2. HeLa control and shSTN1 cells were labeled with EdU for 30 min, pre-extracted to remove soluble MCM, stained for MCM6 and EdU, and then separated by flow cytometry for cell cycle analysis (Fig S2A and B). The cells were then gated to separate G1, S, and G2/M populations (Fig S2C–E and see the Materials and Methods section). This analysis revealed no observable change in the number or intensity of EdU-positive cells with STN1 depletion compared with controls, suggesting bulk DNA synthesis is not significantly affected with STN1 depletion in HeLa cells, as previously reported (Fig 2B) (Wang et al, 2012, 2014). Likewise, when we compared chromatin-bound MCM across the cell cycle, we found the expected increase in MCM-positive cells in G1 (origin licensing) followed by a linear decrease throughout S-phase (origin firing/DNA synthesis) with very few MCM-positive cells in G2/M (Fig 2C)

(Matson et al, 2017). However, the intensity of MCM-positive cells in the G1 population of shSTN1 cells was significantly increased compared with the controls, suggesting increased origin licensing after STN1 depletion (Fig 2C and D). There was also an increase in MCM intensity in S-phase and G2/M (Fig 2E). However, very few MCM-positive cells were within the G2/M population, which was unchanged with STN1 depletion (Fig 2E, MCM-positive cells in G2/M: shNT = 1.5%, shSTN1 = 1.5%, and shSTN1-Res = 1.4%). These data argue against a defect in MCM removal after replication termination because this should lead to an increase in the fraction of MCM-positive cells in G2/M. Instead, our results demonstrate that CST suppresses origin licensing in G1. Similar results were observed in HCT116 shSTN1 cells (Fig S3). Overall, these results provide evidence that CST regulates origin licensing (i.e., MCM loading in G1). Furthermore, we find that chromatin-bound MCM increases in S-phase, which is likely caused, at least in part, by excess origin licensing in G1.

### Depletion of CTC1 or TEN1 is not sufficient to increase chromatin-bound MCM

To demonstrate that this phenotype is not caused by long-term changes from stable knockdown, we next examined whether transient siRNA knockdown of STN1 also increased chromatin-bound MCM. We also determined whether CTC1 or TEN1 knockdown increased MCM chromatin association. The cells were treated with siRNA targeting CTC1, STN1, or TEN1, and MCM levels were then assessed in pre-extracted cells by IF and flow cytometry, as described above in Figs 1 and 2. With siRNA depletion of STN1, we observed a similar increase in MCM compared with stable knockdown. However, we were surprised to find that MCM levels were not increased after CTC1 or TEN1 knockdown (Fig S4). These results suggest that CTC1 or TEN1 depletion are not sufficient to increase MCM levels and STN1 is the critical component of CST required to alter MCM chromatin association (see additional details below).

### Altered CST expression leads to cell type–specific changes in S-phase progression

Because changes in origin licensing (i.e., MCM loading) could alter genome replication, we determined whether STN1 depletion or CST overexpression altered cell growth and cell cycle progression. Interestingly, both HCT116 STN1 depleted and HeLa CST-OE cells exhibit decreased cell proliferation (Fig S5). However, we previously showed that HeLa shSTN1 cells do not exhibit growth defects or defects in cell cycle progression (Stewart et al, 2012). To assess cell cycle progression in the HCT116 shSTN1 and HeLa CST-OE cells, we synchronized the cells by double thymidine block and released them into S-phase. The shSTN1 cells progressed more quickly through S-phase (Fig S6A). In contrast, a minor delay in S-phase progression was observed in CST-OE cells (Fig S6B). Although the changes in S-phase progression cannot be directly attributed to the changes in origin licensing, this does fit with increased or decreased MCM chromatin association altering origin licensing and activation after STN1 depletion or CST-OE, respectively. However, such effects on cell cycle progression and growth may reflect cell type–specific differences (e.g., p53 status, cellular MCM, or CST

levels) or relate to the level of STN1 knockdown, as HeLa shSTN1 cells do not exhibit accelerated S-phase progression or growth defects (Stewart et al, 2012).

### CST interacts with the MCM complex

Because CST affects origin licensing, we next tested whether CST interacts with MCM (Fig 3). First, co-immunoprecipitation (co-IP) experiments were performed using whole cell lysates from cells expressing CST subunits. HA-tagged CTC1, Flag-tagged STN1, or untagged TEN1 were transiently expressed in combination or individually. The lysates were nuclease-treated before IP to ensure that the interaction was not DNA-dependent. Association of MCM subunits or CDC45 with CST was then assessed by Western blot. MCM4 and MCM7 co-immunoprecipitated with CTC1 or STN1 (Fig 3A). CDC45, a component of the CMG complex, was also detected. Interaction with MCM was confirmed by reciprocal co-IP, where endogenous MCM7 was pulled down and STN1 detected (Fig 3B). Co-IP was also performed after HU treatment to see if interactions between CST and MCM or CDC45 were increased upon HU-induced fork stalling. However, neither interaction was significantly altered in response to HU treatment (Fig S7A and B).

Next, yeast-two-hybrid analysis was performed to identify which subunit(s) of MCM and CST interact. Human CTC1-, STN1-, or TEN1-coding regions were fused with the *GAL4* DNA-binding domain (BD), whereas human MCM subunits or CDC45 were linked to the *GAL4* DNA-activation domain (AD). Combinations of the constructs were then transformed into yeast, selected, and spotted onto double synthetic dropout (DDO) medium, lacking histidine and tryptophan, or quadruple synthetic dropout (QDO) medium, lacking tryptophan, leucine, adenine, and histidine. The DDO media was used to select for plasmid transformations and QDO media for cells producing adenine and histidine, which indicates protein interaction. Western blot of whole cell lysate was also performed to confirm the expression of MCM subunits, CDC45, and CST subunits (Fig 3C).

Among the different MCM subunits, we identified strong interactions between STN1 with MCM4 and MCM7 (Fig 3C). A weak interaction was also observed between CTC1 and MCM4 on the less stringent triple synthetic dropout (TDO) medium (Fig S7C). TEN1 did not interact with any MCM subunit. Together, our results identify an interaction between STN1 and the MCM4–MCM7 interface and are consistent with STN1 depletion, but not CTC1 or TEN1, causing increased MCM loading (Fig S2). Interestingly, MCM4–MCM7 sit opposite to the MCM2–MCM5 gate, suggesting that CST would not block opening of the gate or binding of CDC45 and GINS (Riera et al, 2017; Zhai et al, 2017b). We also tested CDC45 and CST subunits in the yeast-two-hybrid assay and did not observe any growth on the TDO or QDO media, which suggests that their association, detected by co-IP, may be bridged by MCM (Fig 3C). Alternatively, multiple CST subunits could be required for interaction, as is the case for CST interaction with pol α (Bhattacharjee et al, 2017; Feng et al, 2018). Overall, our results demonstrate that CST interacts with MCM, with strong interaction between STN1 and MCM4–MCM7. Based on the yeast-two-hybrid data, we propose that this interaction is direct. However, it is possible that an evolutionary-conserved protein could bridge the interaction.

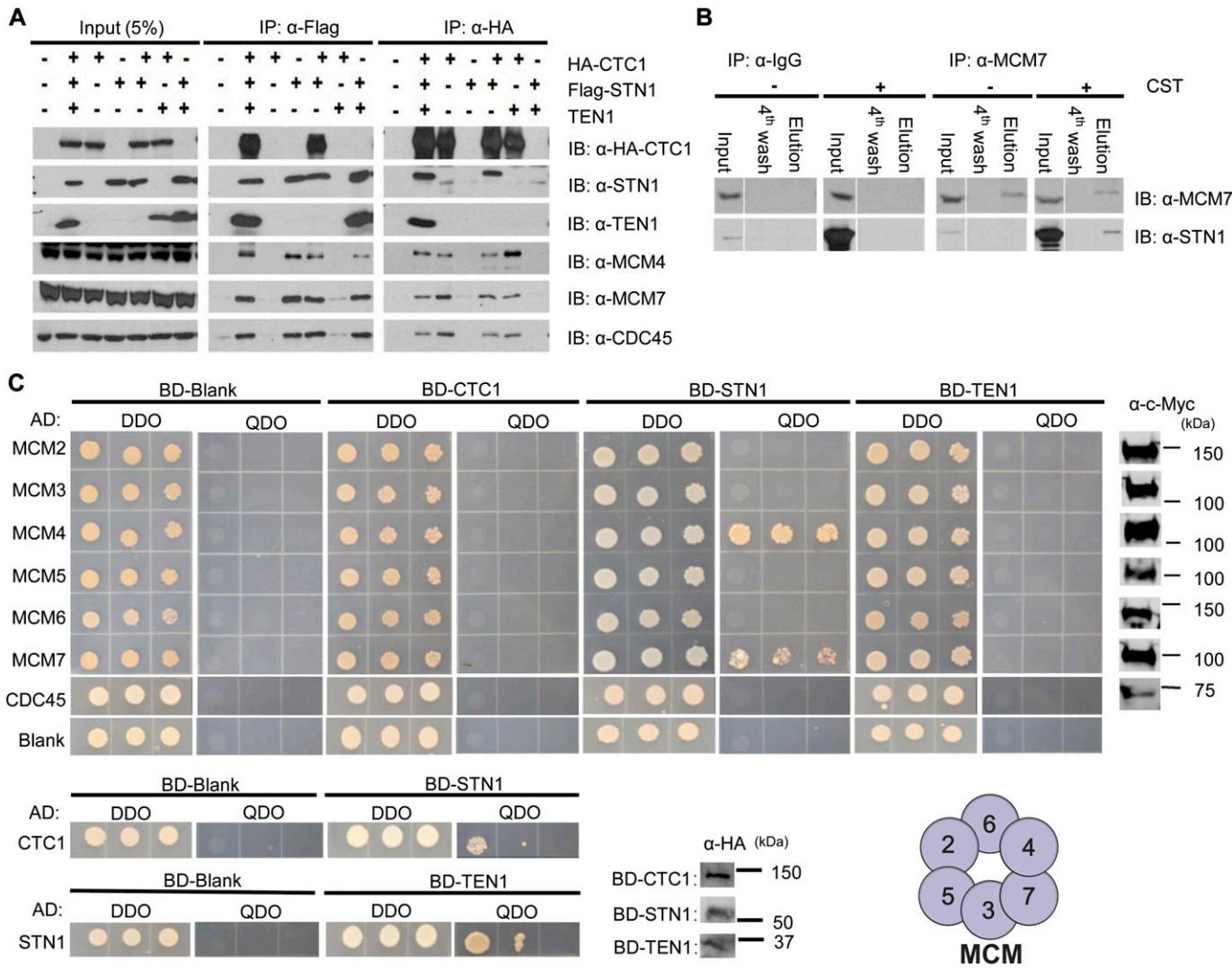

**Figure 3. CST and MCM physically interact.**
**(A)** Co-IP of nuclease-treated lysates from either mock or transfected HEK 293T cells with either Flag or HA antibody, as indicated. Data are representative of three independent biological experiments. **(B)** Reciprocal co-IP experiment with MCM7 or an IgG control antibody of lysates from HEK 293T cells transfected with plasmids expressing CST. Data are representative of three independent biological experiments. **(C)** Yeast-two-hybrid analysis of the interaction between CST and MCM. DDO medium was used for plasmid selection and QDO medium to assess interaction. Colony growth was recorded 3–4 d after plating. Interactions between CTC1 and STN1, and STN1 and TEN1 were used as a positive control (bottom). Data are representative of three independent biological experiments. Expression of MCM subunits and CDC45 was detected by Western blot using a Myc antibody (far right). Expression of CST subunits was detected by Western blot using HA antibody (bottom middle). Representation of MCM hexamer with numbers to indicate subunits (bottom right).

## CST disrupts the interaction between MCM and CDT1

We next sought to determine whether CST binding to MCM was the underlying cause of decreased origin licensing by CST. As mentioned previously, CDT1 plays an essential role in the recruitment and loading of MCM. Because CST interacts with MCM, we hypothesized that binding of CST could obstruct or destabilize CDT1 binding to MCM. Recent biochemical analysis of budding yeast MCM and Cdt1 showed that Cdt1 interacts with MCM2 and MCM6 and weakly with MCM4 (Frigola et al, 2017). Human CDT1 has also been shown to interact with MCM6 (Liu et al, 2012). To more closely examine the interaction between MCM and CDT1 in humans, we performed yeast-two-hybrid analysis (Fig 4A). In

agreement with previous studies, we observed a strong interaction between CDT1 and MCM6. In addition, we find a weaker interaction between CDT1 and MCM4, similar to budding yeast. These findings suggest that CST and CDT1 have overlapping binding surfaces (i.e., CST binds MCM4–MCM7, whereas CDT1 binds MCM6–MCM4).

To test whether CST disrupts CDT1 binding to MCM, CDT1-HA, Flag-MCM subunits, and/or CST were expressed in HEK 293T cells. Flag-tagged MCM2, MCM4, or MCM6 was then immunoprecipitated. The levels of CDT1-HA associated with MCM were then measured with or without expression of the entire CST complex (+CST) or just CTC1 and STN1 (+CS) (Figs 4B and S4). As expected, CDT1 associated with MCM4 and MCM6 in the control (no CST or CS). Association was also

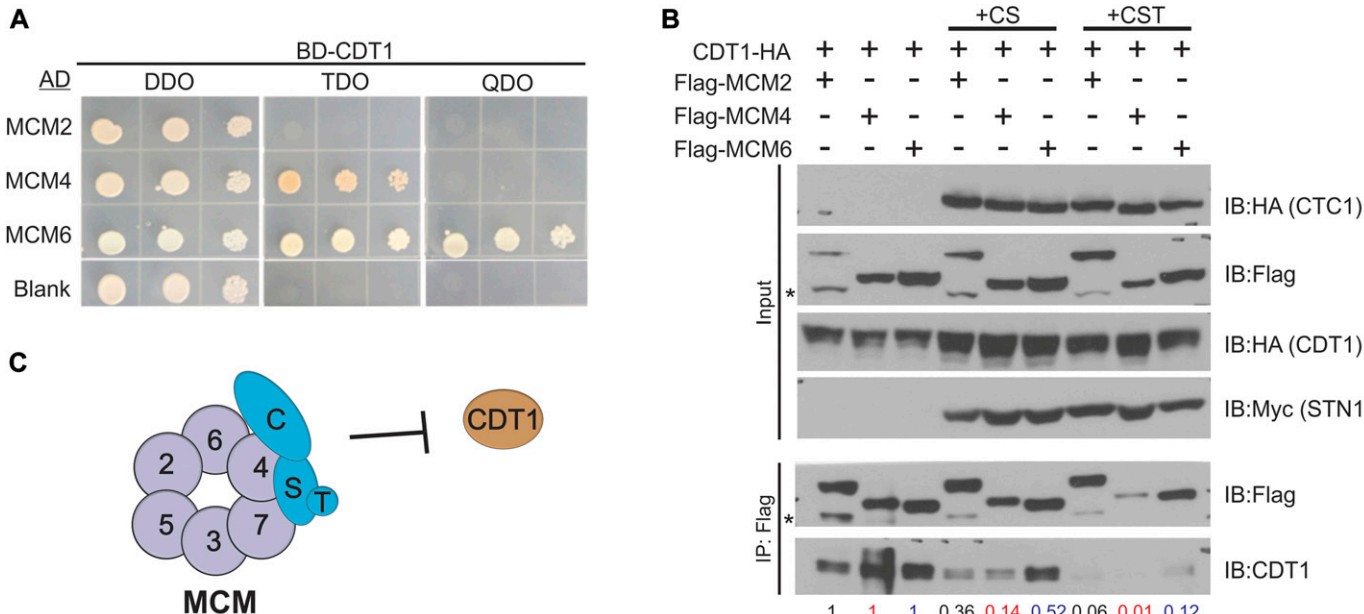

**Figure 4. CST disrupts the interaction between MCM and CDT1.**
**(A)** Yeast-two-hybrid analysis was used to analyze the interaction between CDT1 and MCM. Yeast diploids grown on DDO medium were selected on TDO medium or QDO medium for binding and expression. Colony growth was monitored 3–4 d after plating and incubation. Data are representative of three independent biological experiments. **(B)** Co-IP of lysates from HEK 293T cells expressing Flag-MCM subunits and CDT1-HA with or without expression of CST subunits (+CS: CTC1 and STN1; +CST: CTC1, STN1, and TEN1), as indicated. Relative CDT1-HA band intensity was compared between no CST or CST expression, as indicated below the gel (see the Material and Methods section). Data are representative of three independent biological experiments for +CST. Replicate experiment is shown in Fig S4. **(C)** Model of how CST binding to MCM could block CDT1 binding to MCM.

observed with MCM2. However, when CST was co-expressed, we observed a decrease in CDT1 binding to MCM, suggesting that CST disrupts MCM–CDT1 interaction.

One caveat was that expression of MCM subunits and CDT1 varied to some extent across samples. To account for these differences, we normalize to the amount of MCM pulled down in +CST or +CS samples to MCM IPs without CST expression (see the Materials and Methods section for more details). Furthermore, we normalized the levels of CDT1 in the MCM IP to the CDT1 input levels. Relative CDT1 levels were then quantified in the MCM IP +CST or +CS compared with MCM IP samples without CST expression (Figs 4B and S8, numbers below blot). Here, we observed a significant decrease in CDT1 binding to MCM in the presence of CST. Interestingly, when only CTC1 and STN1 were expressed, the decrease in CDT1 binding to MCM, although still significant, was not as severe as the expression of the entire CST complex. This indicates that binding of TEN1, which likely alters the confirmation of CTC1 and STN1, also contributes to the inhibition of CDT1 binding to MCM. These striking findings indicate that CST disrupts the interaction between MCM and CDT1. However, we do note that depletion of CTC1 or TEN1 did not increase origin licensing (Fig S4). This could be due to incomplete knockdown of CTC1 or TEN1. We propose that blockage of CDT1 occurs through binding of CST to MCM, which prevents/obstructs stable binding of CDT1 (Fig 4C). Disruption of the MCM-CDT1 interaction would directly affect origin licensing (i.e., MCM loading) by preventing MCM recruitment, thus providing a possible explanation for why CST decreases origin licensing.

### STN1 chromatin association increases in S-phase

While examining the levels of chromatin-bound MCM, we also determined whether CST chromatin association changed throughout the cell cycle. HeLa cells stably expressing Flag-STN1 were synchronized by double-thymidine block and collected 0, 1.5, 3, 6, 9, 12, and 24 h after release. Flow cytometry was used to verify cell synchronization and EdU incorporation to identify cells in S-phase (Fig 5A). Western blot analysis for STN1 was then performed on chromatin fractions (Fig 5B). At the 1.5–6-h time points, the cells were predominately in S-phase and transitioned to G2/M by 9 h. Although a small fraction of STN1 remains chromatin-bound throughout the cell cycle, STN1 levels significantly increased on the chromatin at the 1.5–6 h timepoints (Fig 5B), suggesting that CST is recruited to the chromatin during S-phase. Significantly, the timing of this increase differs from the timing of increased CST telomere association, which occurs in the late S/G2 (Chen et al, 2012).

### CST interacts with AND-1 and promotes AND-1 and pol α chromatin binding

Previous studies have shown that CST interacts with pol α and depletion of CST subunits leads to anaphase bridges and chromosome fragility in the absence of exogenous replication stress (Goulian et al, 1990; Stewart et al, 2012; Wang et al, 2012; Chen et al, 2013; Chastain et al, 2016; Wang & Chai, 2018). These observations suggest CST plays additional roles in DNA replication beyond origin licensing. Given that CST is recruited to the chromatin in

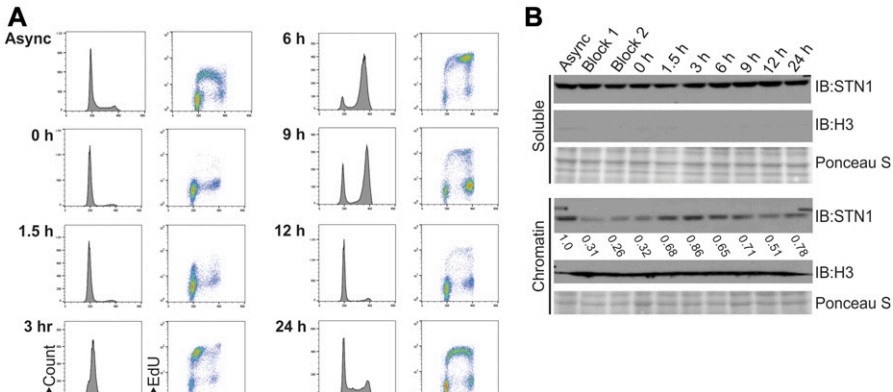

**Figure 5. STN1 chromatin association increases in S-phase.**
**(A)** Cell synchronization by double-thymidine block was confirmed by flow cytometry. Graphs on the left of each timepoint show the DNA content (DAPI) versus number of cells (count) and on the right is the DNA content versus EdU signal intensity. **(B)** Western blot analysis showing soluble and chromatin-associated STN1 throughout the cell cycle. Async: asynchronous cells; block 1: cells after first thymidine block; block 2: cells before second thymidine block. Ponceau S stains total protein and was used as a loading control. Histone H3 was used as a control for chromatin fractionation. Numbers below the STN1 chromatin blot indicate the relative levels of STN1 throughout the cell cycle compared with the asynchronous control. Data are representative of three independent biological experiments.

S-phase and interacts with pol α, we considered whether CST promotes replisome assembly and origin firing, particularly whether CST affects the association of AND-1 and pol α with the replisome.

Before origin firing, pol α is coupled to the replisome by AND-1 (Simon et al, 2014). Because CST also physically and functionally interacts with pol α, we hypothesized that CST may assist or replace AND-1 to link pol α to the replisome under certain situations, such as at G4s or other GC-rich DNA, which are enriched at origins. AND-1, also known as Ctf4 and WDHD1, is a homotrimeric complex composed of conserved WD40 repeats and a SepB domain (Guan et al, 2017). Human AND-1 was recently shown to bind to ssDNA in vitro, which may help position pol α for primer synthesis (Kilkenny et al, 2017).

We first tested whether depletion of STN1 altered chromatin-bound AND-1 in HeLa cells. The cells were pre-extracted, and chromatin-associated AND1 was measured by IF or flow cytometry (Figs 6A and B, S9A and B). STN1 knockdown caused a significant decrease in AND-1 levels, suggesting that CST promotes AND-1 chromatin binding. To ensure that CST had not decreased total cellular AND-1, we checked AND-1 levels in whole cell lysates, where AND-1 was slightly increased in the shSTN1 and shSTN1-Res compared with shNT cells (Fig S9C). Although the reason for these changes are not clear, they indicate that decreased AND-1 chromatin association is not due to decreased cellular AND-1 levels. We also examined AND-1 and pol α in HCT116 shSTN1 cells and found a decrease in both AND-1 and pol α chromatin association (Fig 6C). These findings fit with a role for CST in replisome assembly, as AND-1 association is needed for pre-IC formation. However, it is interesting to note that only a fraction of AND-1 is lost from the chromatin, suggesting that CST is only required at subset of replication origins to recruit AND-1.

To address the possibility that CST indirectly affects AND-1 levels, we next determined whether AND-1 and CST physically associate. Co-IP was performed in cells expressing Flag-STN1, HA-CTC1, or the entire CST complex. We found that STN1 alone or in complex with CST interact with AND-1 (Fig 6D). These results provide evidence that CST directly aids in the loading of AND-1/pol α on the chromatin and hence with the replisome.

Given our previous finding that CST promotes dormant origin firing in response to HU treatment (Stewart et al, 2012; Wang et al, 2014), we next asked whether the dependence of AND-1 on CST for its chromatin association could explain how CST promotes dormant origin firing in response to genome-wide replication fork stalling. Although HU inhibits global origin firing, it promotes the firing of dormant replication origins nearby stalled forks (Ge & Blow, 2010). We hypothesized that CST might enable firing of these dormant origins by recruiting AND-1/pol α. A prediction of this hypothesis is that STN1 knockdown would decrease AND-1 association with dormant origins adjacent to forks that stalled in response to HU treatment. If correct, then this would cause an additional decrease in chromatin-bound AND-1 on top of the decrease already observed after STN1 knockdown in the absence of HU treatment. Thus, the effect of STN1 knockdown on AND-1 chromatin association should be much larger in HU-treated cells.

To test whether fork stalling affected AND-1 levels on the chromatin, the cells were treated with HU for 2 h and IF for chromatin-associated AND-1 was performed (Fig 6B). With HU treatment, we observed decreased AND-1 chromatin association across all cell lines, suggesting that HU-induced replication fork stalling causes an overall decrease in AND-1 chromatin association. This overall decrease in AND-1 is likely due to the repression of global origin firing. When STN1-depleted cells were treated with HU, we observed an additional decrease in AND-1 chromatin association relative to control cells. However, the magnitude of the decrease was the same in the HU-treated and untreated cells (~15%). Because the AND-1 levels were not magnified with STN1 knockdown after HU treatment, this finding indicates that CST is unnecessary for AND-1 to associate with dormant origins that are fired in response to genome-wide replication fork stalling. We, therefore, infer that CST associates with AND-1 and pol α to stabilize replisome formation at a subset of origins that are active during an unperturbed cell cycle.

# Discussion

In this study, we show that CST regulates two different aspects of DNA replication. First, CST interacts with the MCM complex and disrupts its interaction with CDT1, leading to the suppression of

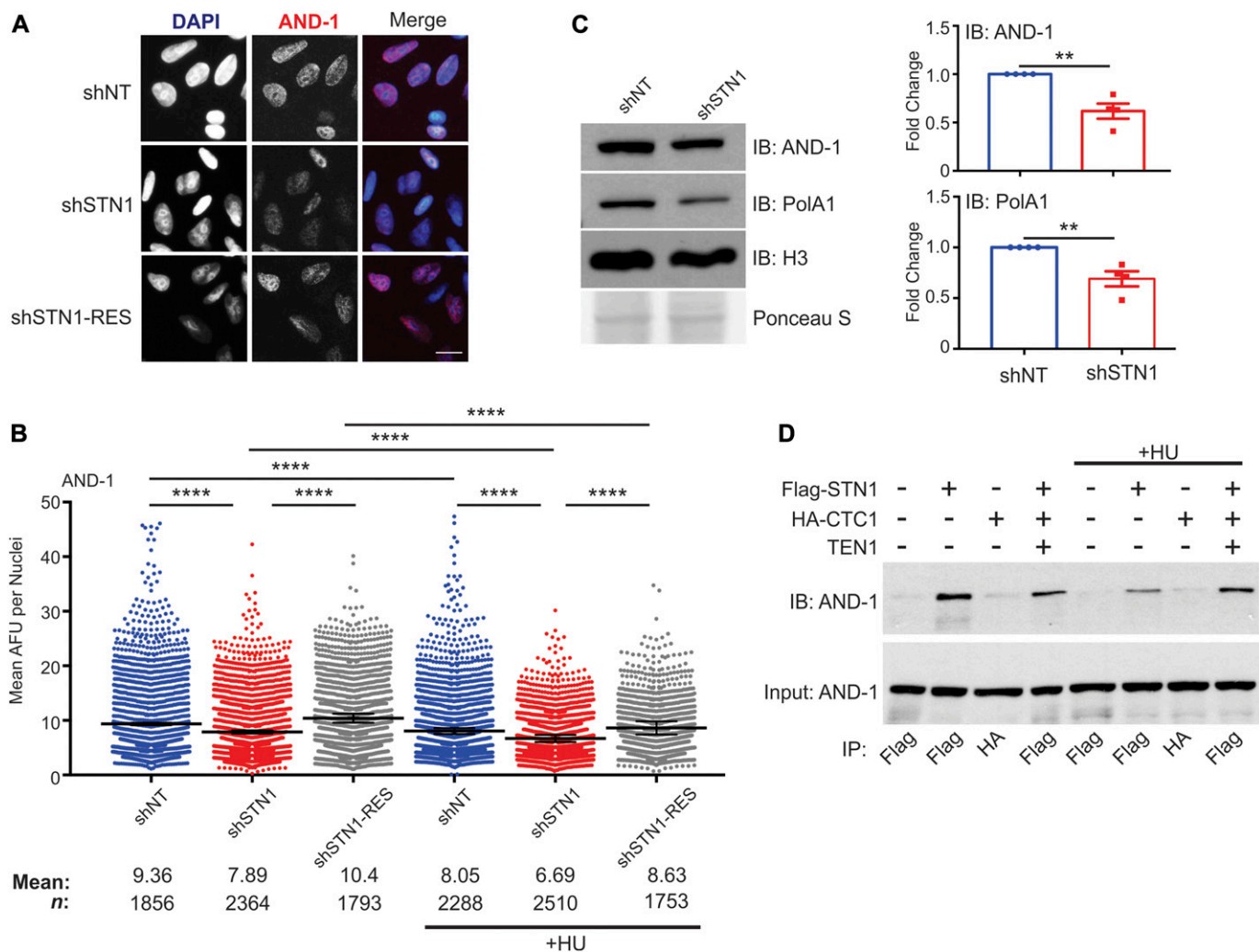

**Figure 6. CST is required for AND-1 and pol α chromatin association.**
**(A)** Representative images of pre-extracted HeLa cells used to measure chromatin-associated AND-1. DAPI: blue, AND-1: red. Scale bar = 12.5 μm. **(B)** Dot plots of mean AND-1 intensity per nuclei in AFU for each cell line, as indicated. Black line and numbers below the graph indicate the mean AFU. Error bars indicate the ±SEM of three independent biological experiments. **(C)** Western blot analysis showing chromatin fractions from HCT116 cells. Ponceau S and histone H3 were used as loading controls. AND-1 and pol α levels were normalized to H3 levels and then normalized to the shNT control. **(D)** Co-IP was performed with Flag or HA antibody in cell lysates from HEK 293T cells, as indicated. 5% input was loaded as a control. For +HU samples in (B) and (D), HU was added 2 h before collection. Data are representative of three independent biological experiments. *n* indicates the number of total nuclei scored. *P*-values were calculated by an unpaired, two-tailed Mann–Whitney test in (B) and *t* test in (C) (****$P \leq 0.0001$, **$P \leq 0.01$).

origin licensing. Second, CST associates with AND-1 to promote AND-1 and pol α chromatin association, presumably with the replisome. It is striking that CST functions in two quite separate aspects of DNA replication, but this is, perhaps, not surprising given the RPA-like nature of CST and that RPA also functions in multiple aspects of DNA replication and repair (Fanning et al, 2006). Moreover, CST has already been shown to play distinct roles in several aspects of telomere replication (Wang et al, 2012; Stewart et al, 2018). It is interesting that the newly discovered roles of CST in origin licensing and replisome assembly are independent of global replication fork stalling, as this suggests that dormant origin firing and RAD51 recruitment after genome-wide fork stalling are distinct CST activities (Stewart et al, 2012; Chastain et al, 2016; Wang & Chai, 2018). At present, the mechanism by which CST facilitates

replication restart after genome-wide fork stalling and whether it is directly or indirectly mediated is unclear. However, here we show direct involvement of CST in origin licensing and replisome assembly.

Origin licensing requires CDT1 interaction with MCM to enable MCM binding to ORC/CDC6 and sequential loading of two MCM hexamers on the chromatin (Ticau et al, 2015, 2017). Our results demonstrate that CST disrupts the interaction between MCM and CDT1 (Fig 4), which may explain how CST suppresses origin licensing (Fig 7A). Prior structural analysis of yeast MCM and OCCM complexes combined with our present data provides insight into the molecular details whereby CST could block the MCM–CDT1 interaction and MCM loading (Abid Ali et al, 2017; Li et al, 2015; Yuan et al, 2016; Yuan et al, 2017; Zhai et al, 2017a; Zhai et al, 2017b). Our yeast-two-hybrid

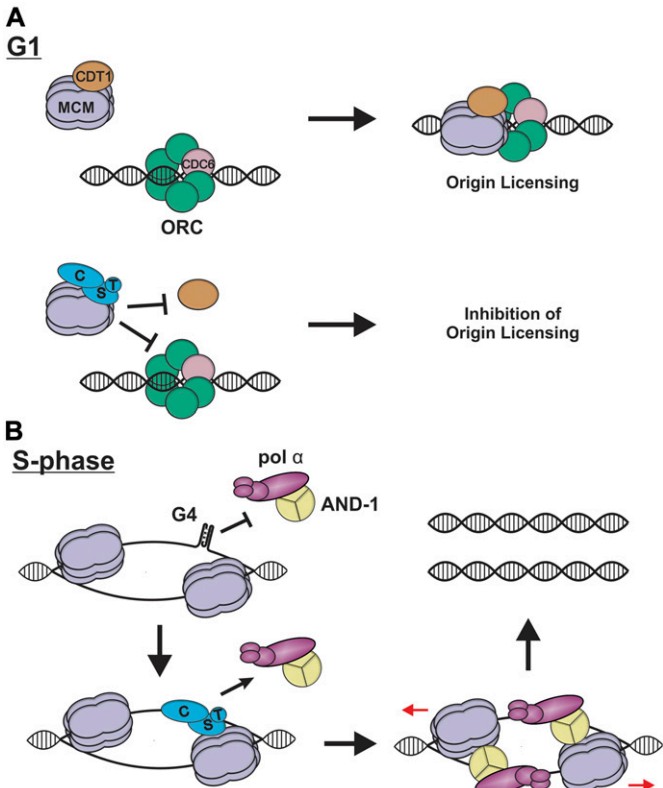

**Figure 7. Potential roles of CST during DNA replication.**
**(A)** In G1, CDT1 interacts with MCM, which facilitates ORC/CDC6 interaction and origin licensing (top). When CST binds to MCM, this prevents CDT1 from interacting with MCM, blocking origin licensing (bottom). **(B)** During S-phase, CST may prevent or remove DNA secondary structures, such as G4s, which inhibit AND-1 from associating with the replisome, only MCM is shown for simplicity. After removal of such structures, AND-1 and pol α associate with CST, leading to replisome assembly and initiation of DNA synthesis.

results indicate that STN1 strongly interacts with the MCM4–MCM7 interface, with weak interactions between CTC1–MCM4 (Fig 3). Within the MCM hexamer, MCM4–MCM7 are located opposite of the MCM2–MCM5 gate used for loading. Recent work suggests that, in yeast, CDT1 interacts with the MCM2–MCM6–MCM4 interface and stabilizes an open ring conformation for MCM loading (Frigola et al, 2017). In addition, NMR studies showed that the C-terminal domain of MCM6 interacts with CDT1, suggesting a similar binding interface in human cells (Wei et al, 2010; Liu et al, 2012). Finally, we find that human CDT1 also interacts with MCM4 (Fig 4A).

Binding of CST to MCM4, which is adjacent to MCM6, could destabilize/block the interaction between CDT1 and MCM (see Fig 3C), leading to a closed ring conformation and the inability for MCM to interact with ORC/CDC6 (Fig 7A). Alternatively, CST binding could cause a conformational shift that prevents CDT1 binding to MCM. Disruption of the MCM–CDT1 interaction may be used to prevent the loading of malformed or incomplete MCM hexamers at origins, or overloading of MCM at specific regions of the genome (e.g., G4s), which could affect replication timing (Das et al, 2015). In budding yeast, CDC6 is known to prevent unproductive MCM loading after OCCM formation (Frigola et al, 2013). CST could provide an additional quality control mechanism to prevent unproductive MCM.

Further biochemical studies are needed to determine the mechanism by which CST or STN1 prevents origin licensing and to understand the contribution of each CST subunit, as CTC1 and TEN1 were needed to maximally inhibit the interaction between MCM and CDT1 (Fig 4), but depletion of either CTC1 or TEN1 did not increase origin licensing (Fig S4).

Altered origin licensing can have a profound impact on genome stability, particularly when combined with defects in other pre-RC or pre-IC factors or oncogene-induced replication stress (Kotsantis et al, 2018). For example, mutations in pre-RC factors, which decrease origin licensing, are known to cause Meier–Gorlin syndrome, a primordial dwarfism disorder (Bicknell et al, 2011a, 2011b). Changes in STN1 mRNA expression were also reported in a number of cancers (Chastain et al, 2016). Our data suggest that STN1 depletion or CST-OE can also alter cell proliferation in HCT116 or HeLa cells, respectively (Fig S4). Interestingly, two Coats plus patients with STN1 mutations had intrauterine growth retardation and cell lines derived from these patients showed decreased proliferation and DNA damage (Simon et al, 2016). Thus, future research to determine whether changes in CST expression or mutation affect origin licensing under different conditions may help clarify the molecular etiology of CST-related diseases. It will also be essential to understand the extent to which defects in origin licensing are counterbalanced with other CST-related functions, such as dormant origin activation after replication stress.

We also find that CST interacts with AND-1 (Fig 5). AND-1 is important for recruitment of pol α, and may also act as a hub for the localization of other proteins to the replisome (Zhu et al, 2007; Im et al, 2009; Yoshizawa-Sugata & Masai, 2009; Hao et al, 2015). In addition, AND-1 has ssDNA-binding activity, which may help position pol α for replication (Kilkenny et al, 2017). A recent report demonstrated that auxin-induced degradation of AND-1, in DT40 cells, caused large stretches of ssDNA, DNA damage, and G2 arrest but, surprisingly, AND-1 degradation did not prevent the initiation of DNA synthesis (Abe et al, 2018). The ability of AND-1–deficient cells to still initiate replication may reflect a role for other factors, such as CST and MCM10, in linking pol α to the replisome. However, without AND-1, uncoupling of pol α from the replisome is likely to be common, leading to ssDNA gaps and incomplete replication (Zhu et al, 2007; Im et al, 2009).

Because G4s are enriched at replication origins (Valton & Prioleau, 2016), we propose that the interaction between CST and AND-1 may be necessary at specific regions of the genome, such as G4s or GC-rich DNA, which could block AND-1 binding. Here, we envision that CST interacts with MCM and removes DNA secondary structures that may form upon initial unwinding but before DNA synthesis (Fig 7B) (Bhattacharjee et al, 2017). CST could then recruit/stabilize AND-1 and pol α to the replisome followed by CST dissociation. This role for CST would be akin to the RPA "hand-off" mechanism, where RPA guides the recruitment of specific proteins to the DNA, which then trade places with RPA through sequentially dissociation of its OB-folds (Fanning et al, 2006; Chen & Wold, 2014). AND-1 would then position pol α to initiate DNA synthesis. A similar situation could occur at replication forks stalled by G4s or other secondary structures. Here, CST would resolve the block and then recruit AND-1/pol α to reinitiate DNA synthesis.

With regard to the timing of CST association with the replisome, studies to detect replisome-associated proteins have failed to detect CST at replisomes actively synthesizing DNA (Miyake et al, 2009; Sirbu et al, 2013). Instead, we propose that CST interacts with CMG before the initiation of DNA synthesis. In support of this idea, our data demonstrated that CST associates with CDC45 (Fig 3). Because CST interacts with MCM and pol α, it is also possible that CST may substitute for AND-1 under certain conditions, as discussed above. It is formally possible that CST could also assist AND-1 with fork restart, fork protection, or DNA damage signaling (Hao et al, 2015; Chen et al, 2017; Li et al, 2017; Abe et al, 2018). However, these scenarios seem unlikely because HU-induced fork stalling did not increase the interaction between CST and AND-1 or further decrease AND-1 levels with STN1 knockdown (Fig 6).

Defects in the recruitment of AND-1/pol α to the replisome could explain the increased anaphase bridges and chromosome fragility seen after depletion or deletion of CST subunits even though bulk, genome-wide DNA replication (as assayed by EdU uptake) remains unaffected in HeLa cells (Stewart et al, 2012; Wang et al, 2012; Chastain et al, 2016; Wang & Chai, 2018). The ~15% decrease in chromatin-bound AND-1 in STN1-depleted cells suggests that CST is required for AND-1 association at only a subset of origins. Replication blocks at these origins (e.g., G-rich regions) could prevent origin firing, leading to anaphase bridges or chromosome fragility without affecting bulk DNA replication. This would fit with CST being a specialized, as opposed to, a general replication factor. In this case, binding of CST, rather than RPA, may be advantageous because it could resolve the replication block without activating ATR and a subsequent DNA damage response.

Overall, our findings provide evidence that CST functions in two distinct aspects of genome-wide DNA replication, namely, origin licensing and replisome assembly. CST was previously shown to interact with pol α and stimulate its activity. Here, we demonstrate that CST interacts with additional replisome components, MCM and AND-1. Future work to untangle how CST expression affects origin licensing, replisome assembly and replication dynamics will help clarify the non-telomeric roles of CST. Furthermore, determining whether CTC1 or STN1 mutations, arising in Coats plus patients, affect origin licensing and firing through interactions with AND-1, and MCM may help decipher the molecular pathogenesis of this disease.

# Materials and Methods

### Cell culture

HeLa 1.2.11 cells were maintained in RPMI 1640 media, HCT116 in McCoy's 5A media, and HEK 293T and HeLa TetOn in DMEM at 37°C with 5% $CO_2$. All cell lines were supplemented with 10% fetal bovine serum and 1% penicillin/streptomycin. Media for HeLa1.2.11 and HCT116 shRNA knockdown cells also contained puromycin (1 μg/ml; Gibco) to maintain selection (Stewart et al, 2012). Doxycycline (1 μg/ml; Sigma-Aldrich) was added to HeLa TetOn wild-type and CST overexpressing (CST-OE) cells 24 h before collection, as previously described (Wang et al, 2014). The cell lines were regularly checked for mycoplasma contamination. For siRNA experiments, 20 nM ON-TARGETplus siRNA SMART pools (Dharmacon) to CTC1 (L-014585-01), STN1 (L-016208-02), TEN1 (L-187549-00), or non-targeting control (D-001810-10-05) were transfected into cells with Lipofectamine RNAiMAX (Thermo Fisher Scientific).

### Whole cell extraction

Cell pellets were suspended in lysis buffer (20 mM, Tris pH 8.0, 100 mM NaCl, 1 mM $MgCl_2$, 0.1% IGEPAL, 1× protease inhibitors [1 μg/ml pepstatin A, 5 μg/ml leupeptin, 1 μg/ml E64, 2 μg/ml aprotinin, and 5 μg/ml antipain], and 1× phosphatase inhibitors [4 mM β-glycerophosphate, 4 mM sodium vanadate, and 20 mM sodium fluoride]) and incubated on ice for 15 min. The samples were then sonicated three times (10 s on/5 s off) at 40% amplitude and rested on ice for 5–10 min. The extracts were treated with Benzonase (0.0625 U/μl; EMD Millipore) for 1 h on ice followed by centrifugation at 19,000 $g$ at 4°C for 5 min. The supernatant was collected and analyzed by Western blot. Protein concentrations were determined with the BCA assay (Thermo Fisher Scientific).

### Chromatin fractionation

Cell pellets were suspended in lysis buffer (10 mM, Tris pH 7.5, 100 mM NaCl, 1 mM $MgCl_2$, 0.34 M sucrose, 0.1% Triton X, 1 mM DTT, 1× protease inhibitors, and 1× phosphatase inhibitors) and then rotated at 4°C for 30 min, before centrifugation at 19,000 $g$ for 20 min at 4°C. The supernatant was saved as the cytosolic fraction. The pellet was resuspended in soluble nuclear lysis buffer (10 mM Tris, pH 7.5, 100 mM NaCl, 1 mM $MgCl_2$, 0.34 M sucrose, 1 mM DTT, 1× protease inhibitors, and 1× phosphatase inhibitors) using one half of the volume of the lysis buffer used above. The samples were incubated at 37°C for 10 min and centrifuged at 19,000 $g$ at room temperature. The supernatant was removed, and the pellet resuspended in RIPA buffer (50 mM Tris–HCl, pH 8.0, 150 mM NaCl, 1% Triton X-100, 0.1% SDS, 1 mM EDTA, 1 mM DTT, 1× protease inhibitors, and 1× phosphatase inhibitors) followed by sonication twice (10 s on/5 s off) at medium intensity. The extracts were rested on ice for ~5 min and treated with Benzonase (0.0625 U/μl; EMD Millipore) for 1 h on ice. The samples were then centrifuged at 19,000 $g$ at 4°C for 5 min and the supernatant saved as the chromatin fraction. Protein concentrations were determined with the BCA assay and samples analyzed by Western blot.

### Antibodies

Primary: MCM3 (sc-390480; Santa Cruz), MCM4 (A300-125A; Bethyl Laboratories), MCM6 (611622; BD Biosciences), MCM7 (sc-22782; Santa Cruz), CDC45 (sc-55569; Santa Cruz), Actinin (sc-17829; Santa Cruz), OBFC1 (STN1) (ab119263; Abcam), TEN1 (Kasbek et al, 2013), WHDH1 (AND-1) (NBP1-89091; Novus), PolA1 (A302-850A; Bethyl Laboratories), α-Tubulin (T9026; Sigma-Aldrich), H3 (9715; Cell Signaling), HA-tag ([anti-mouse: 2367; Cell Signaling] [anti-rabbit: 3724; Cell Signaling]), Flag-tag ([anti-mouse: F1804; Sigma-Aldrich] [anti-rabbit: PA1984B; Thermo Fisher Scientific]), and Myc-tag ([anti-mouse: 05-724; EMD Millipore] [anti-rabbit: ab1906; Abcam]). Secondary: Thermo Fisher Scientific: anti-rabbit-HRP (32460), anti-mouse-HRP (32430), and

anti-goat-HRP (31402); molecular probes: goat–anti-mouse Alexa Fluor 647 (A21235), goat–anti-rabbit Alexa Fluor 647 (A21244), goat–anti-rabbit Alexa Fluor 594 (A11037), goat–anti-mouse Alexa Fluor 594 (A11032), goat–anti-mouse Alexa Fluor 488 (A11029), and goat–anti-rabbit Alexa Fluor 488 (A11034).

## Western blot analysis

15–30 $\mu$g of protein, unless otherwise indicated, were run by SDS–PAGE and transferred to a nitrocellulose membrane. All membranes were checked with Ponceau S staining for transfer efficiency and then blocked in 5% non-fat milk in PBS plus 0.1% Tween 20 (PBST) for at least 2 h. Primary antibodies were diluted in 5% non-fat milk-PBST or PBS and incubated overnight at 4°C. Primary antibodies were removed and the membranes washed 3× for 10 min each in PBST. Secondary antibodies were diluted in 5% non-fat milk-PBST and incubated for at least 2 h at RT. After incubation, the membranes were washed 3× for 10 min each in PBST. The blots were then developed with Western Lightning Plus ECL (Perkin Elmer) or ECL Prime (GE Healthcare).

## IF

Cells were plated onto coverslips and allowed to grow to 50–70% confluency. They were incubated with 50 $\mu$M EdU for 30 min, where indicated. For MCM subunits, soluble proteins were pre-extracted with ice-cold 1× CSK buffer (10 mM Hepes, pH 7.4, 0.3 M sucrose, 100 mM NaCl, 3 mM MgCl$_2$, 1× protease inhibitors, and 1× phosphatase inhibitors) containing 0.1% Triton X-100 for 2–3 min at RT. The cells were then fixed with ice-cold 100% methanol at –20°C for 10 min. The cells were blocked in a 2% BSA/1% Fish Gelatin-PBS solution for at least 1 h at RT or overnight at 4°C. Primary antibodies were diluted in 2% BSA/1% Fish Gelatin-PBS at 1:500 and incubated with the coverslips for at least 1 h followed by three PBST washes. Secondary antibodies were incubated at 1:1,000 with the coverslips for at least 1 h at RT followed by three PBST washes. For AND-1, the cells were pre-extracted and fixed as previously described (Chen et al, 2017). Coverslips were then blocked with 3% BSA in 1× PBS followed by incubation with 1:100 $\alpha$-WDHD1 (AND1) for 1 h at RT. After three PBST washes, the coverslips were incubated with 1:1,000 goat $\alpha$-rabbit Alexa Fluor 594 for 30 min at RT and then washed three times with PBST. Where indicated, EdU was detected per the manufacturer's instructions (Life Technologies). All coverslips were then dehydrated using an ethanol series and mounted with FluoroGel (Electron Microscopy Sciences) containing 0.2 $\mu$g/ml DAPI. Images were taken under 40 × or 60× on an EVOS FL microscope (Thermo Fisher Scientific). The nuclear signal intensity was analyzed in ImageJ as previously described (Stewart et al, 2012).

## Cell synchronization

Cells were plated into 10-cm dishes at ~5 × 10$^5$ cells/ml and allowed to grow overnight. 2 mM thymidine was added and plates were incubated for 14 h at 37°C. After 14 h, media was removed and all cells were washed three times with warm 1× PBS. Fresh media was added and the cells were released for 9 h. After 9 h, a second thymidine block was initiated by the addition of 2 mM thymidine.

The plates were incubated at 37°C for 16 h. After 16 h, the cells were washed three times with warm 1× PBS, and the media replaced. The samples were then collected at specific timepoints (0, 1.5, 3, 6, 9, 12, and 24 h release). After collection, the cells were divided for flow cytometry (see below) or fractionation and Western analysis (see above).

## Flow cytometry

After the cells were collected, the supernatant was removed and MCM6 samples were pre-extracted to remove soluble proteins by the addition of 500 $\mu$l of fresh 1× CSK buffer plus 0.1% Triton X-100 and incubated at RT for 5 min. Leaving the CSK buffer on the cells, 5 ml of ice-cold 100% methanol was then added to the tubes dropwise with gentle vortexing. The tubes were capped, inverted once, and then placed at –20°C for 10 min. The samples were inverted once during incubation to prevent clumping. 5 ml of filter-sterilized 1% BSA-PBS was then directly added to the CSK buffer/100% methanol mixture and the samples were centrifuged at 1,000 $g$ for 5 min. The supernatant was removed and 5 ml of fresh, filter-sterilized 1% BSA-PBS was added to the resuspended cell pellet. For AND-1, the cells were pre-extracted and fixed as previously described, except in solution (500 $\mu$l) (Chen et al, 2017). The samples were then centrifuged at 1,000 $g$ for 5 min. The supernatant was removed and 5 ml of fresh, filter-sterilized 1% BSA-PBS was added to the resuspended cell pellet. The samples were stored at 4°C.

Cells were then spun down at 1,000 $g$ for 5 min. The supernatant was removed and 200 $\mu$l of MCM6 (1:500) or AND-1 (1:100) antibody diluted in 1% BSA-PBST was added to the resuspended cell pellet for at least 1 h at RT, with mild vortexing halfway through the incubation. 5 ml of 1% BSA-PBST was then added and samples spun down at 1,000 $g$ for 5 min and the supernatant removed. A second wash of 5 ml 1% BSA-PBST wash was then performed. The cells were then resuspended and incubated in 250 $\mu$l of goat $\alpha$-mouse Alexa Fluor 647 (1:500) in 1% BSA-PBST for at least 1 h at RT, protected from light, with mild vortexing halfway through the incubation. The cells were washed twice as described above with 5 ml of 1% BSA-PBST.

EdU was detected by Click-iT chemistry. Reaction cocktail was made according to the manufacturer's instructions (Thermo Fisher Scientific). 250 $\mu$l of the complete Click-iT reaction cocktail was added and the samples incubated at RT for 30 min 5 ml of 1% BSA-PBST was then added to each tube and the samples spun down at 1,000 $g$ for 5 min, supernatant removed, and the cells resuspended in the residual liquid. 1 ml of fresh DAPI staining solution (200 $\mu$l 10% Triton X-100, 20 $\mu$l 1 mg/ml DAPI, and 200 $\mu$l 10 mg/ml RNase, into 20 ml 1% BSA-PBS) was added, and the cells were incubated for at least 15 min at RT. The samples were spun at 50 $g$ for 30 s to remove cell clumps and debris through filter-capped tubes (Corning) and run on a BD LSR II Flow Cytometer in the Microscopy and Flow Cytometry Facility at the University of South Carolina, College of Pharmacy. Measurements and analysis were then performed for chromatin-bound MCM6, EdU, and DAPI using FlowJo (FlowJo, LLC). Gates for EdU or MCM-positive cells were created using control samples lacking EdU (Fig S2B). To separate out G1, S, and G2/M populations in Figs 2E and S3D, the following gating was performed (see Fig S2). G1 cells were EdU– cells at ~200 DAPI peak (2n). S-phase cells were EdU+ cells in the ~200–400 DAPI range (2n–4n). G2/M were

EdU– cells in the ~400 DAPI range (4n), which would omit any cells entering G2 during the 30-min EdU labeling period. MCM6+ cells were then selected, using control cells without MCM6 antibody.

### co-IP assay

For co-IP experiments, the cells were grown overnight, such that cells were at 50–80% confluency on the day of transfection. 10 µg of total DNA was mixed with 20 µl polyethylenimine (10 mg/ml) (Polysciences) for each transfection. Plasmids used for transfection include pcDNA3.1-Flag-STN1 (Stewart et al, 2012), pcDNA3.1-HA-CTC1 (Surovtseva et al, 2009), pTRE2-TEN1 (Wang et al, 2014), pInducer20-Blast-CDT1-HA (a gift from Jean Cook, Addgene plasmid #109335), and pcDNA3.1 Flag-tagged MCM subunits (ORF cDNA clones from GenScript). After 48 h, the cells were collected and resuspended in 500 µl lysis buffer (20 mM, Tris pH 8.0, 100 mM NaCl, 1 mM MgCl$_2$, 0.1% IGEPAL, and 1× protease and 1× phosphatase inhibitors), incubated on ice for 15 min and then treated with Benzonase for 1 h at 4°C with rotation to digest the DNA. After incubation, the samples were spun down at 19,000 $g$ for 7 min at 4°C and the supernatant moved to a new tube. The lysates were pre-cleared by adding 50 µl of Protein A/G bead slurry (Santa Cruz) and incubating at 4°C for 30–60 min with rotation. The beads were pelleted at 1,000 $g$ for 1 min at 4°C and the supernatant collected. For IP, 30 µl of 50% M2 $\alpha$-Flag (Sigma-Aldrich) or $\alpha$-HA (Sigma-Aldrich) agarose bead slurry was added to 500 µg pre-cleared whole cell lysate in a total volume of 300 µl and incubated overnight at 4°C with rotation. The beads were then pelleted by centrifugation at 1,000 $g$ for 1 min at 4°C and were washed four times with 1 ml of lysis buffer at 4°C for 5 min with rotation for each wash. Proteins were released from the beads by adding 75 µl 2x sample buffer (100 mM Tris–HCl, pH 6.8, 4% SDS, 20% glycerol, and 0.02% bromophenol blue). The samples were boiled for 5 min before SDS–PAGE and Western blot analysis. For quantification of the co-IP results in Figs 4B and S8, the relative level of MCM pulled down with CST expression was normalized to the samples without CST expression [MCM(+CST)/MCM(–CST) = relative IP levels]. CDT1 in the input were then normalized to CDT1 input levels [CDT1(IP)/CDT1 (input) = CDT1 level]. CDT1 levels were then divided by the relative MCM IP. For example, [(CDT1(IP)/CDT1(input))/(MCM2(+CST) IP/MCM2(–CST) IP) = relative CDT1 in MCM2 IP]. We also quantified CDT1 association to MCM without normalizing for CDT1 input levels and also observed a significant decrease (>50%) in CDT1 association with MCM subunits after CST expression.

### Yeast-two-hybrid screens

The human CTC1, STN1, TEN1, MCM2, MCM3, MCM4, MCM5, MCM6, MCM7, or CDC45 were amplified by PCR using Phusion Polymerase (Thermo Fisher Scientific) from pcDNA3.1 plasmids encoding the cDNA of each protein (MCM plasmids from GenScript). The PCR products were then cloned into the pGBKT7-BD or pGADT7-AD plasmids. NdeI/SalI sites were used to link CTC1, STN1, and TEN1 to pGBKT7-BD, whereas NdeI/XhoI sites were used for pGADT7-AD. For MCM2-7 and CDC45, EcoRI/NdeI sites were used for both plasmids. Mating was performed using *Saccharomyces cerevisiae*

strains Y187 and AH109 transformed with pGBKT7 plasmids or pGADT7 plasmids, respectively. The pGADT7-blank or pGBKT7-blank plasmids were used as negative controls. Healthy diploids (only large [2–3 mm], fresh [<2-mo-old] colonies) on DDO plates, lacking leucine and tryptophan, were selected and cultured in 5 ml YPDA medium overnight at 30°C with shaking, according to the high-stringency selection protocol. Overnight yeast cultures were diluted to OD600 = 1.0, 0.1, and 0.01, from left to right and spotted on DDO, TDO, lacking tryptophan, histidine, and leucine, or QDO plates, lacking adenine, histidine, leucine, and tryptophan. Incubation was performed at 30°C for up to 10 d. Yeast transformation, mating, interaction test, and plasmid isolation were performed using the Yeast Protocols Handbook and Matchmaker GAL4TM Two-hybrid System 3 & Libraries User Manual (Clontech).

### Yeast protein extraction

Yeast cells from the selective media plates were cultured in liquid DDO medium overnight at 30°C. 3 ml yeast culture was spun down and the supernatant discarded. The pellet was resuspended with 150 µl of 2 M LiAc, mixed thoroughly, and incubated on ice for 5 min. The samples were then spun down at 850 $g$ for 5 min at 4°C and the supernatant discarded. The pellet was then resuspended in 100 µl 2× sample buffer and boiled for 5 min. The samples were spun again and the supernatant transferred to a new tube. The supernatant, containing yeast whole protein, was loaded onto an SDS–PAGE gel and analyzed by Western blot.

## Supplementary Information

## Acknowledgements

We would like to thank Carolyn Price, Michael Wyatt, Alan Waldman, Feng Wang, Anukana Bhattachajree, and members of the Stewart lab for critical reading of the manuscript and helpful discussions. We would also like to thank Ji'Vone Freeman and Jazmine Benjamin for assistance in performing experiments. This study utilized the services of the Flow Cytometry Core Facility of the Centers of Biomedical Research Excellence (COBRE) Center for Targeted Therapeutics, supported by National Institutes of Health (NIH) grant 5P20GM109091, at the University of South Carolina with assistance from Chang-uk Lim. This work was supported by the National Institutes of Health (R00GM104409) and startup funds from the University of South Carolina to JA Stewart. BP Caiello is supported in part by the Magellan Scholar Program and an Honors College Senior Thesis/Project Grant through the University of South Carolina.

### Author Contributions

Y Wang: conceptualization, data curation, methodology, and writing—review and editing.
KS Brady: conceptualization, data curation, and writing—review and editing.

BP Caiello: conceptualization, data curation, and writing—review and editing.

SM Ackerson: data curation and writing—review and editing.

JA Stewart: conceptualization, data curation, formal analysis, supervision, funding acquisition, project administration, and writing—original draft, review, and editing.

**Conflict of Interest Statement**

The authors declare that they have no conflict of interest.

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
