## [Reviewer comments · Life Science Alliance]

Life Science Alliance

Human CST suppresses origin licensing and promotes AND-1/Ctf4 chromatin association

Yilin Wang, Kathryn Brady, Benjamin Caiello, Stephanie Ackerson, and Jason Stewart
DOI: <https://doi.org/10.26508/lsa.201800270>

Corresponding author(s): Jason Stewart, University of South Carolina

Review Timeline:

Submission Date:	2018-12-07
Editorial Decision:	2018-12-08
Revision Received:	2019-03-29
Editorial Decision:	2019-04-01
Revision Received:	2019-04-02
Accepted:	2019-04-03

Scientific Editor: Andrea Leibfried

Transaction Report:

Please note that the manuscript was previously reviewed at another journal and the reports were taken into account in inviting a revision for publication at *Life Science Alliance*.

Referee #1 Review

In the manuscript, the authors showed that CST complex involves in helicase loading through preventing Cdt1 from binding to MCM2-7 complex. If true, this would be an interesting finding. However, the data presented by the authors in the manuscript is far from convincing. More concrete data need to be provided for a firm conclusion. It is premature for publication in this journal in its current form.

Major concerns:

1. Regarding the role of CST complex in origin licensing, the authors use a single approach, immunofluorescence (IF) assay, to measure the levels of chromatin-bound MCM proteins through manipulating the expression of one subunit of CST complex, SNT1. The results obtained from these analyses need to be validated by other approaches. For example, the effect of either STN1 depletion or overexpression on MCM loading at several well characterized origins, such as c-Myc, MCM4, and Lamin B2 replication origin regions, should be examined using MCM ChIP assays in G1 cells. In addition, the levels of MCM proteins on the crude G1 chromatin should also be analyzed and visualized using western blotting to indicate the overall amount of chromatin-bound MCMs.
2. Throughout the analyses, the authors only showed the effect of STN1 knockdown on the levels of MCM proteins. Have the authors ever examined the impact of the depletion of the other two proteins of the CST complex, CTC1, and TEN1?
3. Based on the results from the co-IP and yeast two hybrid assays, the authors claim that CST complex competes with Cdt1 to interact with MCM complex, thus preventing MCM loading. It would be more convincing to the readers if the authors could show a direct interaction between the CST complex and recombinant hMCM2-7 proteins (Boskovic et al 2016 Cell cycle) in vitro, and that this interaction could prevent Cdt1 from interacting with MCM. The domains responsible for the interactions between STN1 and MCM subunits should be mapped. The information could be used for mutant construction aimed at disrupting the corresponding interactions to examine their effects on MCM loading.
4. If high-dosage of CST complex could interrupt the normal MCM loading process, DNA replication initiation should be severely affected in the cells, resulting in a delayed entry into S phase and a lengthened S phase as well. The profile of S phase progression of these affected cells should be examined using FACS with the cells released from G1 synchronization.

Other points:

The introduction part about pre-RC assembly needs to be revised. "OCCM complex, also known as pre-RC" is not correct. Pre-RC refers to the MCM double hexamer assembled at the replication origin. "Following formation of the OCCM, two MCM hexamers are sequentially loaded onto the DNA." is also not correct. In the OCCM, the first MCM complex has already been loaded, encircling dsDNA.

Referee #2 Review

In this paper, Wang et al reported two distinct roles of the CTC1-STN1-TEN1 (CST) complex in general DNA replication process. CST has similar properties with RPA and functions mainly at telomeres. The authors previously showed that CST plays a role in restart of stalled replication forks under DNA replication stress. Here, the authors found that reduction of cellular STN1 caused increase in binding of minichromosome maintenance (MCM) proteins to chromatin using immunofluorescence microscopy. Consistently, overexpression of CST results in decrease of chromatin-bound MCM. They demonstrated that STN1 interacts with MCM4 and MCM6 and that CST caused reduction in the interaction of CDT1 with MCM. These results suggest that CST suppresses pre-RC formation in some regions where CST localizes. Interestingly, CST also interacts with AND-1/Ctf4 that is important for recruitment of DNA polymerase alpha/primase for replisome formation and initiation of Okazaki fragment synthesis. shSTN1 caused significant reduction of chromatin-bound AND-1 and DNA Pol alpha, showing that CST enhances formation/maintenance of replisome, possibly in some regions of chromosome. These findings imply novel roles of CST in unperturbed cell cycle in addition to the role in restart of stalled replication forks under the replication stress.

However, I have two concerns on the consequence and significance of the study. CST plays a role in suppressing pre-RC formation in early stage of initiation, whereas it enhances chromatin binding of AND-1 and DNA Pol alpha increasing replisome. I am a little confused what is the overall consequence of CST in DNA replication. Unfortunately, the authors did not provide sufficient interpretation regarding the relations between these apparently opposite roles of CST in DNA replication. The second concern is the authors did not conduct the analyses on specific regions of chromosomes, despite assuming that CST plays in specific regions including G4s and GC-rich regions. It is quite interesting to determine the regions where CST affects MCM-loading AND-1/Pol alpha loading, respectively, and whether promotion of AND-1/DNA Pol alpha binding occurs in the same region where pre-RC is suppressed. It is also interesting where CST localizes in G1 and S phases, respectively. Lack of evidence and discussion on these points made this work uncertain and less attractive.

Minor comments

1. Fig.1B and others. Authors should specify the colors for merged image, magenta/MCM7, blue/DAPI and green/EdU.
2. Fig.1F. Dot distribution of WT (mean 47.8) differs substantially from that in Fig.3C shNT (mean 36.4). Can the authors present a data set in which controls are consistent with each other.
3. Fig.1F and Fig. EV1D. When CST-OE decreased severely chromatin-bound MCM7 and MCM6, did it affect DNA replication and cell cycle progression?
4. Fig.2C. Authors need to add rationale and justification for the dot line at MCM6 intensity value 2 and for "high intensity MCM6". What happens if intensity above the background similarly to Fig.2B is compared for MCM6?
5. Fig.4B. Expression of CST severely impaired the CDT1-MCM interaction. I wonder if DNA replication is delayed under the conditions.
5. Fig.6C. To justify the comparison, the fold change of IB: H3 should be presented. Alternatively the values should be normalized using IB: H3.

6. Several references including Bhattacharjee et al, Higa et al, are incomplete.

Referee #3 Review

This submission by Wang et al explores the effects of manipulating the CST complex in human cells on two different aspects of DNA replication, origin licensing and the recruitment of a Pol alpha and replication checkpoint recruiter, AND-1. Manipulating the levels of CST has downstream effects on MCM chromatin association that are consistent with CST as a negative regulator of MCM loading. The experiments are well-controlled and rigorous, and the data are generally of very high quality. The IF and FACS analysis of MCM loading are admirably quantitative. In addition, the effect of CST overproduction on the co-immunoprecipitation between (overproduced) Cdt1 and MCM is striking. I have two major concerns however that preclude my support for publication in this journal in its present form:

A) There isn't quite enough substance to the finding that CST inhibits MCM loading to constitute a major advance.

1. What is the biological importance of ~3-fold higher/lower MCM loading in G1? As reported by others, modest changes in MCM loading in G1 don't appear as overt proliferation defects, but they might create some new sensitivities. Is there a cellular phenotype that can connect these CST manipulations on a physiological parameter? For example, does G1 length change? Are cells more or less sensitive to genetic perturbations to licensing (e.g. geminin production, partial depletion of a licensing protein)? Changes in numbers of active forks? the authors should choose how best to demonstrate a downstream consequence of the MCM loading difference.

2. The overexpression and depletion experiments analyze cells that have had long-term changes - shRNA or tet-induced stable expression (for how long?). Can they authors rule out the possibility that the MCM loading changes are indirect effects of cell cycle distribution, a secondary effect of S phase perturbations on subsequent G1 phases, etc?

3. The genetic rescue experiments in Figure 2 are not particularly convincing. In particular, the red and grey signals in Figure 2E look the same whereas red should resemble the blue controls if STN is complementing well. (If this assay were working better, it could be the foundation to test CST mutants.) In addition, the gating of "high" MCM cells is an artificial designation that magnifies the phenotype quantified in 2D; there should be some justification for where this cutoff is placed, and it would be even better to not impose a binary classification on data that are largely continuous. Finally, this figure - while impressive in its technical sophistication - is just one biological replicate.

4. The authors should demonstrate interaction of CST with MCM and interference with Cdt1-MCM

association without the need for the very high expression typically associated with plasmid transfections. Is Cdt1-MCM binding enhanced by CST depletion? Analyzing endogenous proteins will also eliminate the need to normalize for differences in total MCM expression.

5. Is the CST effect on origin licensing separable from its well-known effect at replication forks? For example, is ssDNA binding required for the effect on MCM loading? There is very little molecular dissection of this new phenomenon that could put it in context of overall CST function. The text emphasizes the similarity to RPA, but ssDNA is not thought to be relevant for MCM loading.

B) This study is essentially two smaller projects presented together rather than one complete study. The majority of the novelty and the bulk of the stronger data are the potential role of CST in origin licensing as an addition to its more well-known role at replication forks. The very modest effects of CST on And-1 in Figure 6 and the CST chromatin binding in S phase in Figure 5 are too thin and underdeveloped to warrant inclusion, and they distract from the principal finding.

Minor:

a) The claim that CST "directly" binds MCM is a minor overstatement. One would need to test the complexes in isolation since it is formally possible that an evolutionarily-conserved bridging protein mediates the binding.

b) The authors should report how many cells were analyzed in the dot plots and how many independent biological replicates were performed (co-IPs, FACS, etc.).

c) The interpretation that HU has no effect on binding doesn't necessarily rule out a role for replication stress in chromatin recruitment in Figures 5 and 6. Cells generate endogenous replication stress every S phase. Given that, the S phase chromatin association of CST in Figure 5 could have nothing to do with MCM loading and everything to do with its replication fork function.

d) The yeast spotting assays are spliced images; it would be better to not need that extra manipulation to present the data.

e) The two-hybrid reporters encode genes for adenine and histidine biosynthesis, not GAL4 itself (page 8, top).

f) The general protein stain Ponceau S is mis-spelled on several figures.

December 8, 2018

Re: Life Science Alliance manuscript #LSA-2018-00270-T

Dr. Jason A Stewart
University of South Carolina
Biological Sciences
University of South Carolina
715 Sumter Street
Columbia, SC 29208

Dear Dr. Stewart,

Thank you for transferring your manuscript entitled "Human CST suppresses origin licensing and promotes AND-1/Ctf4 chromatin association" to Life Science Alliance. The manuscript was assessed by expert reviewers at another journal before, and those reviewer reports were transferred to us with your permission.

The reviewers at the other journal appreciated the quality of your work, but thought that the support for a biological significance of the results and for the mechanism of CST competing with CDT1 for MCM binding was not sufficient. Given the interest of the newly found CST gain-/loss-of-function effects particularly on replication licensing, these concerns do not preclude publication in Life Science Alliance, and we would like to invite you to provide a revised version for publication here. We would like to ask you to provide a full point-by-point response to the previously raised concerns and to address all minor/specific concerns of the referees. Certain controls (CTC1/TEN1 knockdown effect (reviewer #1, point 2)) should get added and the concern regarding indirect cell cycle-related effects (reviewer #3, point 2) should get addressed. Ideally (but failure to do so will not preclude publication), a second biological replicate for the assay in Fig. 2 should be provided (reviewer #3, point 3), and the effect of CST knockdown on CDT1-MCM interaction should get tested (reviewer #3, point 4).

The typical timeframe for revisions is three months. Please note that papers are generally considered through only one revision cycle.

Thank you for this interesting contribution to Life Science Alliance. We are looking forward to receiving your revised manuscript.

Sincerely,

- A letter addressing the reviewers' comments point by point.
- An editable version of the final text (.DOC or .DOCX) is needed for copyediting (no PDFs).
- High-resolution figure, supplementary figure and video files uploaded as individual files: See our detailed guidelines for preparing your production-ready images, <http://life-science-alliance.org/authorguide>
- Summary blurb (enter in submission system): A short text summarizing in a single sentence the study (max. 200 characters including spaces). This text is used in conjunction with the titles of papers, hence should be informative and complementary to the title and running title. It should describe the context and significance of the findings for a general readership; it should be written in the present tense and refer to the work in the third person. Author names should not be mentioned.

B. MANUSCRIPT ORGANIZATION AND FORMATTING:

Full guidelines are available on our Instructions for Authors page, <http://life-science-alliance.org/authorguide>

Response to Reviewers Comments

Referee #1:

2. Throughout the analyses, the authors only showed the effect of STN1 knockdown on the levels of MCM proteins. Have the authors ever examined the impact of the depletion of the other two proteins of the CST complex, CTC1, and TEN1?

We sincerely thank the reviewer for this suggestion, which has revealed STN1 as the major contributor of origin licensing suppression. We have now performed this experiment following siRNA knockdown of CTC1, STN1 or TEN1 (Figure S4). Our results for STN1 are reproduced with siRNA, providing further confirmation that STN1 knockdown leads to increased MCM. Interestingly, we find that neither CTC1 or TEN1 knockdown results in increased MCM chromatin association by IF or flow cytometry. These findings are consistent with our yeast-two-hybrid data demonstrating that STN1 and not CTC1 or TEN1 interact with MCM (Figure 4).

We have added the following section "Depletion of CTC1 or TEN1 is not sufficient to increase chromatin-bound MCM" to describe these new results:

"To demonstrate that this phenotype is not caused by long-term changes from stable knockdown, we next examined whether transient siRNA knockdown of STN1 also increased chromatin-bound MCM. We also determined whether CTC1 or TEN1 knockdown increased MCM chromatin association. Cells were treated with siRNA targeting CTC1, STN1 or TEN1 and MCM levels were then assessed in pre-extracted cells by IF and flow cytometry, as described above in Figure 1 and 2. With siRNA depletion of STN1, we observed a similar increase in MCM compared to stable knockdown. However, we were surprised to find that MCM levels were not increased following CTC1 or TEN1 knockdown (Figure S4). These results suggest that CTC1 or TEN1 depletion are not sufficient to increase MCM levels and STN1 is the critical component of CST required to alter MCM chromatin association (see additional details below)."

To address these results in the context of the inhibition of the MCM-CDT1 interaction, we also added the following text to the section "CST disrupts the interaction between MCM and CDT1":

"However, we do note that depletion of CTC1 or TEN1 did not increase origin licensing (Figure S4). This could be due to incomplete knockdown of CTC1 or TEN1. We propose that blockage of CDT1 occurs through binding of CST to MCM, which prevents/obstructs stable binding of CDT1 (Figure 4C). Disruption of the MCM-CDT1 interaction would directly affect origin licensing (i.e. MCM loading) by preventing MCM recruitment, thus providing a possible explanation for why CST decreases origin licensing."

Finally, text was added to the Discussion to address these differences:

"Further biochemical studies are needed to determine the mechanism by which CST or STN1 prevents origin licensing and to understand the contribution of each CST subunit, as CTC1 and TEN1 were needed to maximally inhibit the interaction between MCM and CDT1 (Figure 4) but depletion of either CTC1 or TEN1 did not increase origin licensing (Figure S4)."

Other points:

The introduction part about pre-RC assembly needs to be revised. "OCCM complex, also known as pre-RC" is not correct. Pre-RC refers to the MCM double hexamer assembled at the replication origin. "Following formation of the OCCM, two MCM hexamers are sequentially loaded onto the DNA." is also not correct. In the OCCM, the first MCM complex has already been loaded, encircling dsDNA.

We thank the reviewer for bringing this to our attention and have changed the text, as follows:

“Loading of the first MCM hexamer by ORC and CDC6 leads to formation the ORC-CDC6-CDT1-MCM (OCCM) complex. A second MCM hexamer is then recruited and loaded onto the DNA for origin licensing to form the pre-replication complex (pre-RC).”

Referee #2:

Minor comments

1. Fig.1B and others. Authors should specify the colors for merged image, magenta/MCM7, blue/DAPI and green/EdU.

This has been corrected in the revised manuscript. The colors are now indicated in the figure legends and we have added color coding to the titles above the images in Figure 1B, 1E and 6A.

2. Fig.1F. Dot distribution of WT (mean 47.8) differs substantially from that in Fig.3C shNT (mean 36.4). Can the authors present a data set in which controls are consistent with each other.

We believe that the reviewer is referring to differences in Figure. 1F and Figure 1C and their point is well taken. However, these are two different types (or subclones) of HeLa cells. In Fig 1C, HeLa1.2.11 cells are used, which were originally used in STN1 knockdown studies due to their long telomeres (Stewart et al. 2012, EMBO J). In Figure 3C, HeLa TetOn cells are used (Wang, et al. Cell Cycle, 2014). Based on subtle differences in the subclones, we would not necessarily expect the levels of MCM staining to be identical. This likely explains the differences observed. Also, while exposure time was kept constant for a given cell type (e.g. HeLa1.2.11, HeLa TetOn, HCT116), exposure times between cell types (HeLa1.2.11 vs HeLa TetOn) was not necessarily the same, which could also reflect differences in relative AFU.

3. Fig.1F and Fig. EV1D. When CST-OE decreased severely chromatin-bound MCM7 and MCM6, did it affect DNA replication and cell cycle progression?

We have now included growth curves and cell cycle profiles of synchronized HeLa CST-OE cells as well as HCT116 shSTN1 cells (Figure S5 and S6). We previously reported growth curves and cell synchronization for the HeLa shSTN1 cells, which showed no defects in cell growth or cell cycle progression (Stewart et al. 2012, EMBO J) and have reconfirmed these findings (data not shown). We have added the following section titled “Altered CST expression leads to cell type specific changes in S-phase progression” to the manuscript, which describes these results:

“Since changes in origin licensing (i.e. MCM loading) could alter genome replication, we determined whether STN1 depletion or CST overexpression altered cell growth and cell cycle progression. Interestingly, both HCT116 STN1 depleted and HeLa CST-OE cells exhibit decreased cell proliferation (Figure S5). However, we previously showed that HeLa shSTN1 cells do not exhibit growth defects or defects in cell cycle progression (Stewart et al., 2012). To assess cell cycle progression in the HCT116 shSTN1 and HeLa CST-OE cells, we synchronized the cells by double thymidine block and released them into S-phase. The shSTN1 cells progressed more quickly through S-phase (Figure S6A). In contrast, a minor delay in S-phase progression was observed in CST-OE cells (Figure S6B). While the changes in S-phase progression cannot be directly attributed to the changes in origin licensing, this does fit with increased or decreased MCM chromatin association altering origin licensing and activation following STN1 depletion or CST-OE, respectively. However, such effects on cell cycle progression and growth may reflect cell type specific differences (e.g. p53 status, cellular MCM or CST levels) or relate to the level of STN1 knockdown, as HeLa shSTN1 cells do not exhibit accelerated S-phase progression or growth defects (Stewart et al., 2012).”

The following text was also added to the discussion regarding these results:

“Our data suggest that STN1 depletion or CST-OE can also alter cell proliferation in HCT116 or HeLa cells, respectively (Figure S4). Interestingly, two Coats plus patients with STN1 mutations had intrauterine growth retardation and cell lines derived from these patients showed decreased proliferation and DNA damage (Simon et al., 2016). Thus, future research to determine whether changes in CST expression or mutation affect origin licensing under different conditions may help clarify the molecular etiology of CST-related diseases.”

Furthermore, previous results demonstrated that when the HeLa CST-OE cell line was subjected to exogenous replication stress new replication origins were activated and cell survival was increased compared to wild type cells (Wang et al. Cell Cycle. 2014). We suggest that the role of CST in replication restart may partially compensate for decreased origin licensing following CST-OE and thus lead to only minor changes in cell cycle progression and proliferation. Other groups have also shown that >75% depletion of MCM does not affect replication in the absence of replication stress, Thus, the decrease in origin licensing observed with CST-OE may not reach the threshold required to drastically alter S-phase progression (Ge, et al. Genes Dev. 2007). Future work will focus on untangling how different CST functions affect origin licensing, S-phase progression and DNA synthesis.

4. Fig.2C. Authors need to add rationale and justification for the dot line at MCM6 intensity value 2 and for "high intensity MCM6". What happens if intensity above the background similarly to Fig.2B is compared for MCM6?

This line was originally chosen to highlight increased MCM intensity of the STN1 knockdown samples and later the percentage of cells in this population was quantified. We agree with the reviewer that this is an arbitrary designation and have now removed the line, omitted the graph and replaced it with a new graph, which compares the mean signal intensity of MCM positive cells between cell lines (Figure 2C and S3C). This new analysis shows that there is still a significant increase in the mean intensity of MCM positive cells with STN1 depletion. The text has been altered to describe this new analysis in the section “STN1 depletion leads to increased MCM in G1 and S-phase”:

“However, the intensity of MCM positive cells in the G1 population of shSTN1 cells was significantly increased compared to the controls, suggesting increased origin licensing after STN1 depletion (Figure 2C-D).”

5. Fig.4B. Expression of CST severely impaired the CDT1-MCM interaction. I wonder if DNA replication is delayed under the conditions.

We also considered that DNA replication and cell growth was significantly delayed in cells overexpressing CST. However, we only observe a minor cell growth defect and delay S-phase progression in these cells compared to wild type (see comment for #3).

5. Fig.6C. To justify the comparison, the fold change of IB: H3 should be presented. Alternatively the values should be normalized using IB: H3.

We have now normalized the levels of AND-1 and pol α to H3 in Figure 6C, which shows similar results to normalizing to Ponceau S.

6. Several references including Bhattacharjee et al, Higa et al, are incomplete.

We thank the reviewer for their careful reading of the manuscript. The references have been corrected.

Referee #3:

2. The overexpression and depletion experiments analyze cells that have had long-term changes - shRNA or tet-induced stable expression (for how long?). Can they authors rule out the possibility that the MCM loading changes are indirect effects of cell cycle distribution, a secondary effect of S phase perturbations on subsequent G1 phases, etc?

The stable HeLa shSTN1 cell line is from a single clone so were originally cultured until sufficient cells were obtained for freeze down. The passage number beyond that is maintained to as few as possible with new stocks frozen shortly after unthawing. Profiles of the stable HeLa STN1 knockdown cells have been previously published (Stewart et al. EMBO J. 2012) and showed no significant changes in cell cycle profile. However, the HCT116 shSTN1 cells show growth defects and changes in cell cycle (Figure S4). This cell line was derived from a pool of cells following drug selection. HCT116 cells are also have functional p53, which would affect cell growth under conditions of DNA damage or replication stress. The CST-OE cells were stably selected and TEN1 is under a doxycycline-inducible promoter (Wang et al. Cell Cycle. 2014). CTC1 and STN1 are constitutively expressed but increase significantly when doxycycline is added. Doxycycline is added 24 h prior to each experiment so these higher levels are not present until the day prior to collection. We also have now analyzed cell growth and S-phase progression, which showed defects for HCT116 shSTN1 and HeLa CST-OE cell lines (see Referee #2, comments #2 and Figure S5 and S6).

To more directly address these concerns, we performed transient knockdown of STN1 with siRNA (see Reviewer #1, comment #2) and found a similar increase in chromatin-bound MCM levels compared to stable shRNA knockdown, indicating that increased MCM also arises with transient knockdown of STN1.

3. The genetic rescue experiments in Figure 2 are not particularly convincing. In particular, the red and grey signals in Figure 2E look the same whereas red should resemble the blue controls if STN is complementing well. (If this assay were working better, it could be the foundation to test CST mutants.) In addition, the gating of "high" MCM cells is an artificial designation that magnifies the phenotype quantified in 2D; there should be some justification for where this cutoff is placed, and it would be even better to not impose a binary classification on data that are largely continuous. Finally, this figure - while impressive in its technical sophistication - is just one biological replicate.

The cutoff line and graph for the high intensity MCM cells has been removed and new graphs depicting changes in mean intensity in all MCM positive cells is now included (see also Reviewer #2, #4).

While the shSTN1-RES cells do not fully rescue the phenotype, there is a substantial decrease in the MCM6 positive cells compared to our controls. In fact, analysis the signal intensity of MCM positive cells is similar between shNT and shSTN1-RES cells (Figure 2C). In addition, incomplete rescue of the shSTN1-RES cell line has been seen with other phenotypes in our previous work and may arise from expression levels, influence from the N-terminal Flag-tag or subtle differences between the single clones isolated for the shNT and shSTN1 cell lines (Stewart et al. 2012, EMBO J). We have now replicated these results with siRNA knockdown of STN1 in the HeLa and HCT116 cells (Figure S4), which shows similar results to stable knockdown of STN1. Together with the results from the HCT116 shSTN1 cells, these results provide strong evidence that STN1 depletion increases origin licensing.

The flow cytometry data are a representation of three biological replicates for both the HeLa and HCT116 shSTN1 cells. This is now clearly stated in the figure legend. These findings are also now replicated with transient siRNA knockdown of STN1, in three independent biological replicates (Figure S4).

4. The authors should demonstrate interaction of CST with MCM and interference with Cdt1-MCM association without the need for the very high expression typically associated with plasmid transfections. Is Cdt1-MCM binding enhanced by CST depletion? Analyzing endogenous proteins will also eliminate the need to normalize for differences in total MCM expression.

We made multiple attempts to detect the interaction with endogenous protein in different cell lines with and without STN1 depletion or CST overexpression. However, we were unable to co-IP sufficient levels of endogenous CDT1 or MCM to reliably detect interact, regardless of CST expression levels. We also tried expressing CDT1 in the HeLa or HCT116 shSTN1 cell lines but were unable to reliably express or detect CDT1 at sufficient levels for IP experiments. Due to these technical issues, we are not able to show the IP with endogenous CDT1 or in STN1 depleted cells. In our opinion, the best approach will be to perform *in vitro* binding experiments with recombinant, purified CDT1, MCM and CST. However, while we are pursuing such experiments, we feel that they are beyond the scope of the current study. For this reason, we have changed the text to suggest that disruption of MCM-CDT1 interaction as one possible explanation for the suppression of origin licensing. The following sentence of the section “CST disrupts the interaction between MCM and CDT1”, has been altered to reflect this:

“Disruption of the MCM-CDT1 interaction would directly affect origin licensing (i.e. MCM loading) by preventing MCM recruitment, thus providing a possible explanation for why CST decreases origin licensing.”

Minor:

a) The claim that CST "directly" binds MCM is a minor overstatement. One would need to test the complexes in isolation since it is formally possible that an evolutionarily-conserved bridging protein mediates the binding.

We have added to following sentence to address this possibility and removed the work “directly” from this subsection, as follows:

“Based on the yeast-two-hybrid data, we propose that this interaction is direct. However, it is possible that an evolutionary-conserved protein could bridge the interaction.”

b) The authors should report how many cells were analyzed in the dot plots and how many independent biological replicates were performed (co-IPs, FACS, etc.).

The number of biological replicates performed for each experiment has been added to the figure legends and the number of cells for the dot plots below the x-axis in the figures.

c) The interpretation that HU has no effect on binding doesn't necessarily rule out a role for replication stress in chromatin recruitment in Figures 5 and 6. Cells generate endogenous replication stress every S phase. Given that, the S phase chromatin association of CST in Figure 5 could have nothing to do with MCM loading and everything to do with its replication fork function.

We agree with the reviewer and were trying to make the point that HU does not affect the AND-1 chromatin association. We did not test how HU affected chromatin bound STN1 levels. We have sought to clarify this in the text with the following changes to the section “CST interacts with AND-1 and promotes AND-1 and pol α chromatin binding”:

“Since the AND-1 levels were not magnified with STN1 knockdown following HU treatment, this finding indicates that CST is unnecessary for AND-1 to associate with dormant origins that are fired in response to genome-wide replication fork stalling.”

d) The yeast spotting assays are spliced images; it would be better to not need that extra manipulation to present the data.

All the yeast-two-hybrid assays in original Figure 3C were performed at the same time and two biological replicates had been performed. However, other mutants were also tested so the MCM subunits were not on single plates with each CST subunit. We have re-run the experiment so that all

MCM subunits are plated with each of the CST subunits (Figure 3C). Interestingly, this time we did not observe the weak interaction previously observed between MCM5 and CTC1. Since this interaction appears weak and inconsistent (observed two out of three times) under the most stringent conditions (QDO media), we have removed reference to a potentially weak interaction between MCM5 and STN1 until future studies can confirm their interaction.

e) The two-hybrid reporters encode genes for adenine and histidine biosynthesis, not GAL4 itself (page 8, top).

This error has been corrected in the text. The sentence now reads:

“The DDO media was used to select for plasmid transformations and QDO media for cells producing adenine and histidine, which indicates protein interaction.

f) The general protein stain Ponceau S is mis-spelled on several figures.

We thank the reviewer for identifying these mistakes and the figures have been updated.

April 1, 2019

RE: Life Science Alliance Manuscript #LSA-2018-00270-TR

Dr. Jason A Stewart
University of South Carolina
Biological Sciences
University of South Carolina
715 Sumter Street
Columbia, SC 29208

Dear Dr. Stewart,

Thank you for submitting your revised manuscript entitled "Human CST suppresses origin licensing and promotes AND-1/Ctf4 chromatin association". I appreciate the introduced changes and would be happy to publish your paper in Life Science Alliance pending final revisions necessary to meet our formatting guidelines:

- please note that we only have supplementary figures in LSA, there are still some callouts to EV figures, please fix
- please add scale bars to Fig 1B, 1E, 6A
- please link your ORCID iD to your profile in our submission system, you should have received an email with instructions on how to do so

A. FINAL FILES:

-- Summary blurb (enter in submission system): A short text summarizing in a single sentence the study (max. 200 characters including spaces). This text is used in conjunction with the titles of

papers, hence should be informative and complementary to the title. It should describe the context and significance of the findings for a general readership; it should be written in the present tense and refer to the work in the third person. Author names should not be mentioned.

B. MANUSCRIPT ORGANIZATION AND FORMATTING:

Sincerely,

Andrea Leibfried, PhD
Executive Editor
Life Science Alliance
Meyerohofstr. 1
69117 Heidelberg, Germany
t +49 6221 8891 502
e a.leibfried@life-science-alliance.org
www.life-science-alliance.org

April 3, 2019

RE: Life Science Alliance Manuscript #LSA-2018-00270-TRR

Dr. Jason A Stewart
University of South Carolina
Biological Sciences
University of South Carolina
715 Sumter Street
Columbia, SC 29208

Dear Dr. Stewart,

Thank you for submitting your Research Article entitled "Human CST suppresses origin licensing and promotes AND-1/Ctf4 chromatin association". It is a pleasure to let you know that your manuscript is now accepted for publication in Life Science Alliance. Congratulations on this interesting work.

DISTRIBUTION OF MATERIALS:

Again, congratulations on a very nice paper. I hope you found the review process to be constructive and are pleased with how the manuscript was handled editorially. We look forward to future exciting submissions from your lab.

Sincerely,
